# Cancer microbiota: a focus on tumor-resident bacteria

Gerlanda Vella[1] & Maria Rescigno [ID][1,2][✉]

## Abstract

**Accumulating evidence highlights the presence of an intratumoral microbiota across various cancer types. Among all the microorganisms comprising the tumor-associated microbiota, tumor-resident bacteria (TRB) are increasingly recognized as critical regulators of cancer biology. Within tumor tissues, these microorganisms interact with various components of the tumor microenvironment (TME) and influence both tumor-promoting and tumor-suppressing pathways, underlying their dual role in cancer. Fully understanding the functional roles of TRB and their complex interactions with components of the TME requires the application of multimodal technologies. Developing strategies to modulate TRB —either by eradicating pathogenic populations or harnessing beneficial ones—holds great promise for advancing cancer treatment. In this review, we summarize the most recent insights into TRB. We discuss their possible origins and their implications on cancer biology, focusing on their roles in cancer development, metastasis establishment, immune modulation, and therapy response. Finally, we describe bacteria-based strategies and address the major challenges in detecting and analyzing these microbial communities in tumors.**

**Keywords** Microbiota; Intratumoral Microbiota; Tumor Microenvironment; Metastasis; Therapy Efficacy
**Subject Categories** Cancer; Microbiology, Virology & Host Pathogen Interaction

## Introduction

The human microbiota consists of trillions of microorganisms—including bacteria, fungi, archaea, and viruses—that play significant roles in human health (Cheng et al, 2024; Ferrari and Rescigno, 2023; Gollwitzer et al, 2014; Naik et al, 2012; Sender et al, 2016; Tofani et al, 2024). Historically, microbiota research focused on body sites enriched with microbial communities, such as the gut, oral cavity, and skin. However, the idea that tumors might harbor microorganisms dates back to the 19th century, when bacteria were observed within solid tumors (Cao et al, 2024; Dudgeon and Dunkley, 1907). While these initial observations hinted at a potential role for microbiota in cancer, they were not fully recognized and explored at that time due to limitation of techniques, which could not reliably exclude contamination or detect very low microbial biomass.

Only recently, the developments of next-generation sequencing (NGS) technologies, particularly the 16S rRNA gene sequencing which targets a conserved region of the bacterial ribosomal RNA gene, have enabled the detection and characterization of bacteria at low abundance in challenging environments like solid tumors. In 2020, Nejman and colleagues provided the first in-depth comprehensive characterization of tumor-associated microbiota by analyzing over 1500 tumor tissues alongside their adjacent healthy counterparts. Their study revealed a diverse bacterial community within tumor tissues across multiple cancer types, including lung, breast, brain, melanoma, ovary, and pancreatic tumors (Nejman et al, 2020). Subsequently, two pivotal studies in 2022 expanded the analysis of the tumor microbiota to fungal microorganisms. Independently, Narunsky-Haziza et al and Dohlman et al detected fungi within cancer specimens from a variety of tumor types, such as lung, stomach, and colon cancers (Dohlman et al, 2022; Narunsky-Haziza et al, 2022). These discoveries underscore the complexity of the tumor microenvironment (TME) and significantly broaden our understanding of its composition. The tumor-associated microbiota has now been recognized as an integral part of the TME, contributing to a highly intricate and dynamic environment (Fig. 1).

Most of the current knowledge on the tumor-associated microbiota comes from studies on tumor-resident bacteria (TRB) which are the most extensively investigated microbial populations in cancer research. Recently, a spatial characterization of bacterial distribution within tumor tissue, in relation to immune cells and other components of the TME has been provided. TRB are intracellular, located in the cytosol of both cancer and immune cells (Fu et al, 2022; Nejman et al, 2020). Moreover, bacteria communities reside in highly immunosuppressive microniches as revealed by state-of-the-art in situ spatial-profiling technologies applied to oral squamous cell carcinoma (OSCC) and colorectal cancer (CRC) (Galeano Niño et al, 2022). These regions are characterized by an enrichment of mature CD66b+ myeloid cells and an upregulation of the immunosuppressive molecule arginase 1 and the immune checkpoint protein CTLA-4. Bacterial communities populate microniches that are less vascularized and associated with poorly proliferating cancer cells compared to

[1]IRCCS Humanitas Research Hospital, via Manzoni 56, 20089 Rozzano, Milan, Italy. [2]Department of Biomedical Sciences, Humanitas University, via Rita Levi Montalcini 4, 20072 Pieve Emanuele, Milan, Italy. [✉]E-mail: maria.rescigno@hunimed.eu

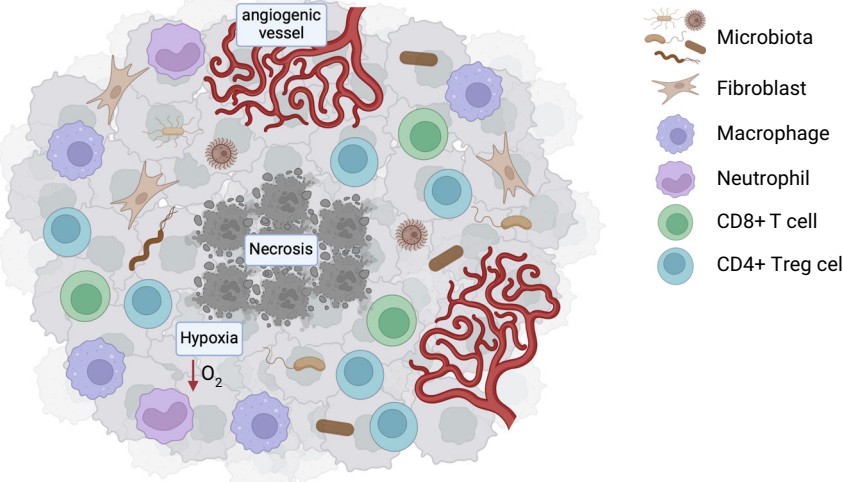

**Figure 1. The complexity of the tumor microenvironment.**

The tumor microenvironment (TME) is a complex and dynamic system. It is characterized by the presence of fibroblasts, poor CD8+ T-cell infiltration, and an enrichment of immunosuppressive regulatory CD4+ T cells (Tregs), tumor-promoting macrophages, and neutrophils. Additionally, the TME typically contains angiogenic tumor vessels that are abnormal, immature, and leaky, leading to increased interstitial pressure and dysfunctional blood flow, which promote hypoxia and necrosis. Together, immunosuppression, low-oxygen levels, and necrosis create ideal conditions for microbiota colonization and survival.

bacteria-negative tumor areas (Galeano Niño et al, 2022). The non-random, spatially organized distribution of TRB within tumors raises intriguing questions about the potential for cross-talk between TRB and other TME components which may influence key aspects of tumor progression and therapy response.

## Origin of tumor-resident bacteria

Despite the recent considerable interest in understanding the biology of the intratumoral microbiota, the origin of TRB remains poorly understood. To date, three putative sources of TRB have been identified: (1) mucosal barriers (Fig. 2A); (2) hematogenous spread (Fig. 2B); (3) tumor-adjacent normal tissue (Fig. 2C).

During tumorigenesis, the integrity of the mucosal barrier may be compromised, allowing bacteria colonizing the mucosal surfaces to invade the TME. Compromised mucosal barriers may serve as a source of TRB, especially for tumors from mucosal sites with external cavity exposure, including lung, pancreatic, colorectal, and cervical cancers (Riquelme et al, 2019; Xue et al, 2023). However, this mechanism is unlikely to account for the presence of microbiota in tumors located at sites distant from mucosal surfaces. Thus, it is conceivable that the origin of microbiota in tumors located in non-mucosal tissues might involve other mechanisms.

Second, given the similar microbial communities found in both tumor tissue and tumor-adjacent normal tissue (Nejman et al, 2020), the latter has been proposed as a potential source of TRB. However, since the origin of bacteria in most healthy organs remains unclear, it is yet to be determined whether microbes spread from healthy tissue to the tumor site, if the process occurs in the opposite direction, or if bacteria residing in healthy tissue expand as the tumor forms and become an integral part of the TME.

Finally, the circulatory system may serve as a conduit for microbes to reach the tumor site through hematogenous spreading.

Bacteria may freely circulate in the bloodstream or may migrate within circulating cancer cells during the metastatic process (Abed et al, 2020; Bertocchi et al, 2021; Fu et al, 2022). Yu and colleagues, elegantly demonstrated that intravenously injected light-emitting bacteria (*Escherichia coli, Vibrio cholerae, Salmonella typhimurium,* and *Listeria monocytogenes*) infiltrated and replicated within primary murine brain and bladder tumors, without causing bacteremia (Yu et al, 2004). The hematogenous spread is facilitated by both passive and active mechanisms. Tumor vessels are typically leaky, disorganized, immature, and are characterized by unsteady blood flow and loosely connected endothelial cells (Vella et al, 2023). Bacteria may passively access tumors through the vascular leakage and be flushed into the tumor due to the strong blood influx (Leschner et al, 2009). Bacterial infiltration can also be actively regulated. One identified active mechanism by which blood circulating bacteria seed into tumors involves the interaction between the bacterial fibroblast activation protein-2 (Fap2) and the D-galactose-β(1-3)-N-acetyl-D-galactosamine (Gal-GalNAc) molecule overexpressed by several malignant tissues, including breast cancer and CRC (Abed et al, 2016; Parhi et al, 2020). *Fusobacterium nucleatum*, a well-studied bacterium in the context of cancer, is a normal inhabitant of the oral cavity (Abed et al, 2020; Segata et al, 2012). Its migration from the mouth to the tumor via the bloodstream is mediated by the galactose-binding lectin Fap2, allowing the bacterium to selectively adhere to CRC and breast cancer cells while sparing healthy tissues with low Gal-GalNAc expression (Abed et al, 2016; Parhi et al, 2020).

Notably, the same bacterium can utilize different routes of migration. For instance, oral *F. nucleatum* may translocate to CRC through the bloodstream during episodes of transient bacteremia, which can occur due to chewing or daily oral hygiene activities (Abed et al, 2020). In addition, *F. nucleatum* can reach CRC by descending through the digestive tract as indicated by studies where mice inoculated with *F. nucleatum* via oral gavage developed more

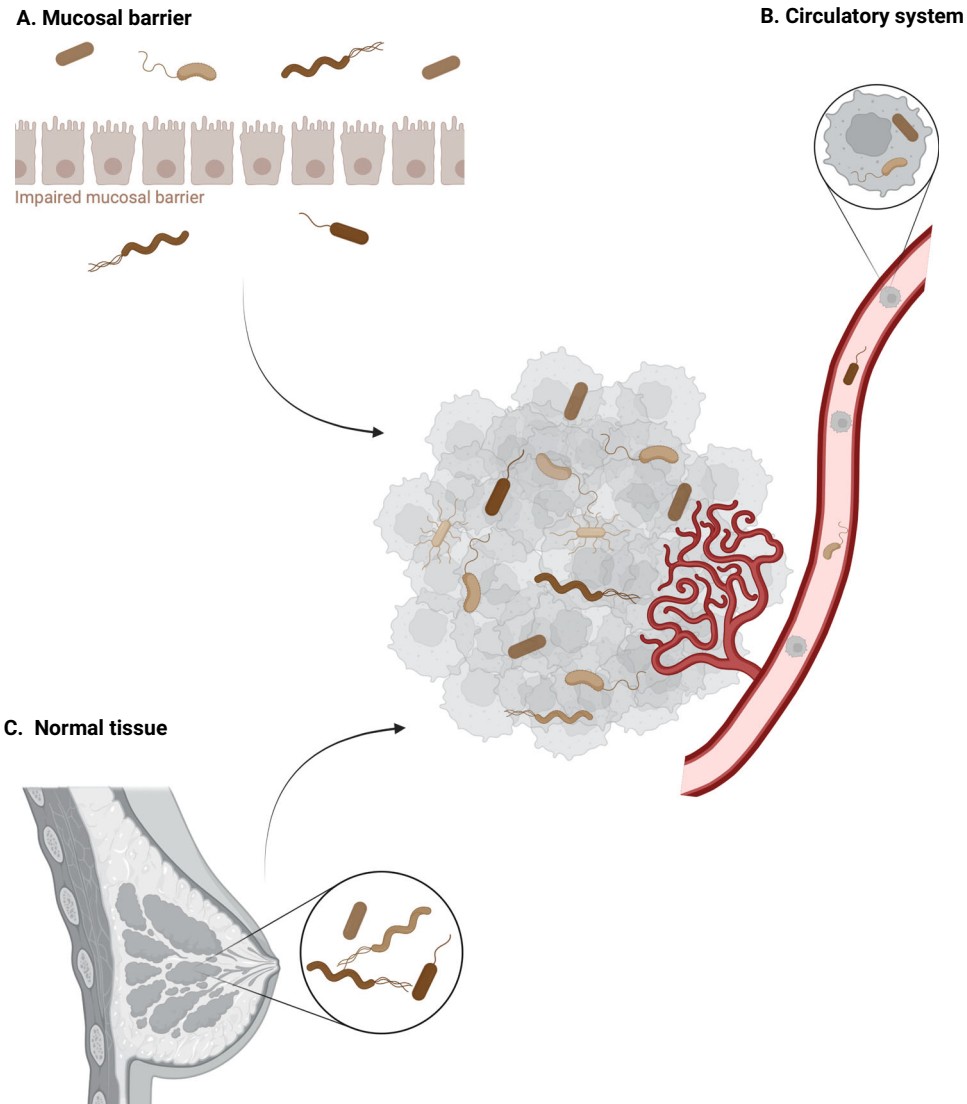

**Figure 2. Putative sources of tumor-resident bacteria.**

(A) Mucosal barriers, if impaired, may lead to the spread of bacteria from the mucosal sites to the TME. (B) Bacteria can be transported by the circulatory system, either freely circulating as a result of bacterial translocation or wound infection or within circulating cancer cells. The leaky and immature neo-angiogenic tumor blood vessels facilitate the hematogenous spread of bacteria within the TME. (C) Healthy tissue-resident bacteria can migrate to adjacent tumor sites or expand during tumorigenesis due to significant changes in the local environment.

colonic tumors compared to mice inoculated with PBS or control bacteria (Kostic et al, 2013; Rubinstein et al, 2019).

To identify the origin of TRB and gain a better understanding of their migration patterns, genomic microbiota analyses of fecal, oral, tumor-adjacent normal tissue, and tumor tissue can be particularly helpful. For instance, a comparison of single-nucleotide variants of representative bacteria isolated from nasopharyngeal carcinoma (NPC) tissues with fecal, oral, and nasopharyngeal microbiota suggested that TRB in NPC may originate from microorganisms residing in the nasopharynx (Qiao et al, 2022). Similarly, comparative 16S rRNA gene sequencing analysis of oral, cancerous pancreatic tissue, and tumor-adjacent normal tissue, revealed a potential migratory route of *Porphyromonas gingivalis* from the oral cavity to pancreatic cancer (PC) tissues (Tan et al, 2022).

Regardless of the bacterial source, an immunosuppressive, nutrient-rich, and hypoxic TME appears to create favorable conditions for bacterial colonization and persistence (Battaglia et al, 2024; Galeano Niño et al, 2022; Shi et al, 2020) (Fig. 1). Facultative and obligate anaerobic bacteria, which grow in oxygen-deprived or low-oxygen environments, are particularly drawn to necrotic and hypoxic tumor regions. Necrosis, in particular, provides a reservoir of nutrients and protection from host immune responses. In addition, lactate and amino acids produced and released by cancer cells may foster bacterial colonization and survival within the TME (Nong et al, 2023; Senevirathne et al, 2025). Although pathogens are typically cleared by phagocytic immune cells through macropinocytosis and subsequent lysosomal degradation (Wong et al, 2017), bacteria have evolved a range of

**Table 1. TRB regulation of tumor initiation and progression.**

| Bacteria | Cancer type | Mechanism/signaling involved | Effect | Ref. |
|---|---|---|---|---|
| *F. nucleatum* | CRC | Bacterial FadA binds to E-Cadherin on cancer cells and, through ANXA1, activates the Wnt/β-catenin signaling pathway. Bacterial formate activates the AhR pathway, promoting cancer stemness and Th17 cell infiltration. | Pro-tumoral | Meng et al, 2021; Rubinstein et al, 2019; Rubinstein et al, 2013; Ternes et al, 2022 |
| *B. fragilis* | CRC | Activation of NF-κB signaling in both immune and colonic epithelial cells | Pro-tumoral | Chung et al, 2018; Dejea et al, 2018 |
| *S. typhimurium* | CRC | Activation of Wnt-β-catenin pathway through the type-III secretion system effector AvrA | Pro-tumoral | Lu et al, 2012; Lu et al, 2014 |
| *E. coli* | CRC | Bacterial colibactin alters genomic stability by inducing DNA adduct formation and double-strand breaks | Pro-tumoral | Nougayrède et al, 2006; Wilson et al, 2019 |
| *H. pylori* | GC | Bacterial oncoprotein CagA promotes cell stemness and induces chronic inflammation via activation of the Wnt-β-catenin and NF-κB pathways | Pro-tumoral | Javaheri et al, 2016; Königer et al, 2016; Suzuki et al, 2015 |
| *S. anginosus* | GC | Binding of bacterial TMPC with the host ANXA2 receptor activates the MAPK pathway | Pro-tumoral | Fu et al, 2024 |
| *P. gingivalis* | PC | Infiltration of TANs and secretion of NE by TANs | Pro-tumoral | Tan et al, 2022 |

CRC colorectal cancer, GC gastric cancer, PC pancreatic cancer, FadA Fusobacterium adhesin A, ANXA1 Annexin A1, AhR aryl hydrocarbon receptor, TMPC transmembrane porin C, ANXA2 Annexin A2, TAN tumor-associated neutrophil, NE neutrophil elastase.

sophisticated strategies to circumvent the immune system during tumor colonization. To avoid phagocytosis, bacteria translocate effector proteins that manipulate the host cell cytoskeleton, preventing their uptake by phagocytes. In cases of successful internalization, bacteria can evade bacteriolysis by disrupting the endosomal membrane or interfering with endosomal trafficking and phagosome maturation, allowing their survival inside the host cell (Black and Bliska, 2000; Chakravortty et al, 2002; Eriksson et al, 2000; Hornef et al, 2002). Moreover, some of the bacterial species form biofilm, which consists of complex microbial communities enclosed in self-produced polymeric matrix, facilitating their protection to the immune system (Chessa et al, 2016; Drusano et al, 2011; Tomkovich et al, 2019). Even if these microbial survival mechanisms require further studies in the cancer setting, it is conceivable that the same or similar mechanisms are used by bacteria to hijack the immune response during the hematogenous spread and cancer colonization.

## Tumor-resident bacteria regulate carcinogenesis

An increasing body of research provides evidence for the involvement of bacteria in tumor initiation and progression (Table 1). The oncogenic potential of bacteria is primarily mediated by their ability to induce chronic inflammation, alter the TME, cause DNA damage, and disrupt key signaling pathways. One prominent example is *F. nucleatum*, which has been found to be abundant in several tumors, including CRC, breast, and pancreatic cancer (Castellarin et al, 2012; Li et al, 2023; Mima et al, 2015; Mitsuhashi et al, 2015; Parhi et al, 2020). Increased levels of this bacterium in CRC have been correlated with more advanced disease stage and poorer clinical outcome, suggesting that *F. nucleatum* may serve as a prognostic biomarker in CRC patients (Castellarin et al, 2012; Kostic et al, 2012; Mima et al, 2016; Yamamoto et al, 2021). Mechanistically, *F. nucleatum* promotes CRC progression via the amyloid-like *Fusobacterium* adhesin A (FadA), which binds to E-cadherin (CDH1) on cancer cells, leading to Wnt/β-catenin signaling activation and modulation of inflammatory and oncogenic responses. The formation of the FadA–E-cadherin–β-catenin complex is mediated by Annexin A1 (ANXA1), a β-catenin modulator. Upon binding to CRC cells, FadA further elevates ANXA1 expression in a positive feedback loop, exacerbating CRC progression (Meng et al, 2021; Rubinstein et al, 2019; Rubinstein et al, 2013). Additionally, *F. nucleatum* can enhance CRC stemness and invasiveness through its metabolic product, formate, which activates the aryl hydrocarbon receptor (AhR) pathway and promotes the infiltration of Th17 cells (Ternes et al, 2022). *Helicobacter pylori*, classified as a type I carcinogen, is another bacterium strongly associated with cancer development, particularly in gastric cancer (GC) (Mitani et al, 2004; Uemura et al, 2001). The first step in *H. pylori* pathogenesis involves the binding of the bacterial adhesin HopQ to gastric epithelial cells through specific cellular carcinoembryonic antigen-related cell adhesion molecules (CEACAMs). This allows the translocation of its virulence factor CagA into the cytoplasm of host cells via the type IV secretion system. Within host cells, CagA disrupts various intracellular signaling pathways involved in cell proliferation and apoptosis, and promotes a pro-inflammatory microenvironment conducive to

oncogenesis (Javaheri et al, 2016; Königer et al, 2016; Suzuki et al, 2015; Takahashi-Kanemitsu et al, 2020). *Streptococcus anginosus* has recently emerged as an additional bacterium implicated in GC carcinogenesis. The transformation of gastric epithelial cells conducted by *S. anginosus* is mediated by the activation of the mitogen-activated protein kinase (MAPK) pathway, a key signaling cascade in regulating cell proliferation. This activation occurs after bacterial binding and colonization of host cells via the surface protein transmembrane porin C (TMPC), which interacts with the Annexin A2 (ANXA2) receptor on gastric epithelial cells (Fu et al, 2024). Enterotoxigenic *Bacteroides fragilis* (ETBF) is enriched in precancerous lesions, particularly in familial adenomatous polyposis (FAP) patients (Dejea et al, 2018). ETBF fosters chronic inflammation by activating IL-17-dependent Nuclear Factor kappa B (NF-κB) signaling in both immune and colonic epithelial cells, ultimately promoting colonic tumorigenesis (Chung et al, 2018; Dejea et al, 2018; Wu et al, 2009). Similarly, *Salmonella typhimurium* can induce chronic activation of the β-catenin pathway in intestinal cells through the type III secretion system effector AvrA, enhancing cell proliferation and promoting colonic tumorigenesis (Lu et al, 2012; Lu et al, 2014). Furthermore, *Porphyromonas gingivalis* has been associated with pancreatic carcinogenesis, as it has been found to be more abundant in pancreatic cancer (PC) tissues compared to adjacent normal tissues (Tan et al, 2022). Intratumoral *P. gingivalis* promotes infiltration of tumor-associated neutrophils (TAN) and induces the release of neutrophil elastase by TANs, accelerating pancreatic cancer development (Tan et al, 2022).

In addition to modulating immune responses and signaling pathways, tumor-resident bacteria (TRB) can induce DNA damage. This is generally mediated by reactive oxygen species (ROS) or toxins (Irrazabal et al, 2020; Nougayrède et al, 2006; Wilson et al, 2019). Certain strains of *Escherichia coli* harbor the polyketide synthase (pks) pathogenicity island, which encodes a set of enzymes responsible for synthesizing the unstable genotoxin colibactin. This toxin induces DNA adduct formation and double-strand breaks, thereby contributing to genomic instability (Nougayrède et al, 2006; Wilson et al, 2019; Xue et al, 2019). Pks+ *E. coli* promotes a distinctive mutational signature which has been detected in a subset of CRC patients, suggesting a direct causal role of pks+ *E. coli* in colorectal tumorigenesis (Pleguezuelos-Manzano et al, 2020).

These observations indicate that bacteria significantly contribute to cancer development through various mechanisms, playing a multifaceted role in malignant transformation. Identifying microbial contributors to cancer development can offer valuable insights into novel targets for both cancer prevention and treatment. Aspirin, a nonsteroidal anti-inflammatory drug targeting cyclooxygenase-2 (COX-2), is a well-known chemopreventive agent for CRC. Its chemopreventive effects are largely attributed to the ability to suppress key pathways involved in inflammation and tumorigenesis, including the inhibition of COX-2-mediated prostaglandin production, as well as the NF-κB and Wnt/β-catenin signaling pathways (Chan et al, 2012, 2007). More recently, aspirin's chemopreventive role has also been linked to its capacity to modulate *F. nucleatum* growth and gene expression. In the APC$^{Min/+}$ mouse model of CRC, dietary supplementation with aspirin completely inhibited *F. nucleatum*-driven colonic tumor formation, potentially by reducing the expression of the bacterial pro-tumorigenic adhesins Fap2 and FadA (Brennan et al, 2021). In light of these findings, the question remains whether preventive aspirin use should be recommended for individuals at high risk of CRC. Moreover, the potential of microbial signatures as biomarkers for cancer prognosis needs further investigation.

## Tumor-resident bacteria alter anti-tumoral drugs

Emerging evidence points into the direction that not only microbes residing in the gut (Alexander et al, 2017; Daillère et al, 2016; Iida et al, 2013; Viaud et al, 2013) but also those within the tumor can influence the efficacy and bioavailability of anti-tumoral agents (Colbert et al, 2023; Geller et al, 2017; LaCourse et al, 2022; Wang et al, 2024; Yu et al, 2017; Zhang et al, 2019). These microbial influences can occur through several mechanisms, including direct enzymatic modification of the chemotherapeutic agents, or through alterations in key components of the TME (Colbert et al, 2023; Geller et al, 2017; LaCourse et al, 2022; Wang et al, 2024; Yu et al, 2017; Zhang et al, 2019) (Table 2). For instance, Gammaproteobacteria, which are enriched in pancreatic ductal adenocarcinoma (PDAC), can directly metabolize the chemotherapeutic agent gemcitabine into its inactive form via the enzyme cytidine deaminase (CDD$_L$), mediating chemoresistance in a murine CRC model (Geller et al, 2017). Similarly, *E. coli* isolated from CRC tumors can metabolize 5-fluorouracil (5-FU), a first-line chemotherapeutic agent used in CRC treatment, through the upregulation of genes involved in uracil metabolism, such as *pre*T and *pre*A (responsible for dihydrouracil synthesis) and *psu*G and *psu*K (involved in pseudouridine metabolism) (LaCourse et al, 2022). This bacterial activity not only protects cancer cells from the cytotoxic effects of chemotherapy but also shields susceptible bacterial strains, including *F. nucleatum*, thereby promoting the recurrence of CRC (LaCourse et al, 2022).

Beyond these direct metabolic effects, TRB can also regulate chemotherapy efficacy indirectly. The observation that *F. nucleatum* preferentially binds to CRC cells expressing the β-catenin modulator ANXA1 which has been reported to be overexpressed in 5-FU-resistant colon cancer cells, has suggested a link between *F. nucleatum* infection and chemoresistance (Onozawa et al, 2017; Rubinstein et al, 2019). *F. nucleatum* can induce resistance to 5-FU in CRC cells by upregulating the host's anti-apoptotic factor BIRC3, which supports cancer cell survival during treatment (Zhang et al, 2019). In addition, *F. nucleatum* can indirectly abolish 5-FU and oxaliplatin toxicity, activating cancer autophagy via downregulation of miR-18a and miR-4802 in infected CRC cells (Yu et al, 2017). Similarly, *Lactobacillus iners* can cause radiation resistance and gemcitabine resistance via tumor metabolic rewiring of cervical cancers (Colbert et al, 2023). Although validations in mouse models are lacking, *L. iners*-induced metabolic rewiring might be driven by an increase of bacterial L-lactate in the tumor microenvironment which would make cancer cells "addicted" to lactate, enhancing lactate utilization, and lactate-regulated induction of reactive oxygen species signaling pathways by cancer cells, particularly after radiation (Colbert et al, 2023).

Overall, these observations indicate that TRB plays a critical role in regulating the host response to anti-tumoral drugs. Unraveling the mechanisms by which TRB alter drug efficacy and bioavailability presents valuable opportunities for developing targeted

**Table 2. TRB in cancer therapy.**

| Bacteria | Therapy type | Influence on therapy | Mechanism | Ref. |
|---|---|---|---|---|
| Gammaproteobacteria | Chemotherapy | Negative | Metabolization of gemcitabine into its inactive form via the enzyme cytidine deaminase | Geller et al, 2017 |
| E. coli | Chemotherapy | Negative | Metabolization of 5-FU through upregulation of genes involved in uracil metabolism | LaCourse et al, 2022; Ternes et al, 2022 |
| F. nucleatum | Chemotherapy | Negative | Induction of 5-FU resistance via upregulation of the host's anti-apoptotic factor BIRC3. Reduction of 5-FU and oxaliplatin toxicity via activation of cancer autophagy | Chung et al, 2018; Dejea et al, 2018; Yu et al, 2017; Zhang et al, 2019 |
| | Immunotherapy | Positive | Enhanced response to anti-PD-1 immunotherapy via butyric acid production | Wang et al, 2024 |
| L. iners | Radiotherapy, chemotherapy | Negative | Induction of radio-chemoresistance to gemcitabine mediated by metabolic rewiring of cancer cells | Colbert et al, 2023 |
| Bifidobacterium | Immunotherapy | Positive | Increased efficacy of anti-CD47 immunotherapy through enhanced antigen-presenting capacities of DCs | Shi et al, 2020 |
| L. paracasei | Immunotherapy | Positive | Increased anti-PD-1 efficacy through upregulation of MHC-I expression on cancer cells | Ferrari et al, 2023 |

5-FU 5-fluorouracil, MHC-I major histocompatibility complex class I.

interventions to overcome resistance to chemotherapy and radiotherapy.

## Tumor-resident bacteria modulate immune cells and immunotherapy response

The relationship between microbes and immune cells is quite intricate. Innate immune cells, which serve as the host's primary defense against invading pathogens, recognize microbial components, such as bacteria, via pattern recognition receptors (PRR). This detection initiates inflammatory signaling cascades that subsequently activate adaptive immune cells, facilitating pathogen clearance and the establishment of long-term immunological memory (Li and Wu, 2021).

Within the TME, bacteria can significantly modulate the infiltration, differentiation, and function of immune cells, thereby influencing the tumor growth and tumor's responsiveness to immunotherapy (Bender et al, 2023; Díaz-Basabe et al, 2024; Galeano Niño et al, 2022; Lattanzi et al, 2023; Riquelme et al, 2019; Shi et al, 2020; Tan et al, 2022; Xu et al, 2021) (Fig. 3A,B). Studies on CRC reported that both *F. nucleatum* and *P. gingivalis* favor iNKT cell conversion to a pro-tumoral phenotype, sustaining cancer progression (Díaz-Basabe et al, 2024; Lattanzi et al, 2023). Mechanistically *P. gingivalis* hampers iNKT cytotoxicity by interfering with the iNKT cell lytic machinery through the upregulation of the immune checkpoint chitinase 3-like-1 protein (CHI3L1) (Díaz-Basabe et al, 2024). In vitro, *P. gingivalis*-primed human iNKT cells exhibited reduced expression of cytotoxic molecules, such as granzyme B and perforin, along with impaired lytic degranulation. CHI3L1 blockade using a neutralizing antibody restored their killing activity against human CRC cell lines (Díaz-Basabe et al, 2024). Moreover, another way by which *P. gingivalis* alters tumor immunity is by favoring the recruitment of immunosuppressive TANs via the upregulation of neutrophil-attracting chemokines (e.g., Cxcl1, Cxcl2, and Cxcr2) and elevating neutrophil elastase (NE) secretion in pancreatic cancer (Tan et al, 2022). In addition to its impact on iNKT cells, *F. nucleatum* has been implicated in the recruitment of neutrophils and macrophages within the CRC microenvironment (Galeano Niño et al, 2022; Xu et al, 2021). This bacterium regulates the miR-1322/CCL20 axis in cancer cells through the NF-kB signaling pathway favouring macrophage infiltration and M2 polarization, which are associated with a pro-tumoral and pro-metastatic role (Xu et al, 2021).

Conversely, under certain conditions TRB can also favor immune activation eliciting anti-tumoral responses (Bender et al, 2023; Overacre-Delgoffe et al, 2021; Shi et al, 2020; Wang et al, 2024) (Table 2). One of the mechanisms of immune cell activity modulation is mediated by bacterial–metabolic products, commonly referred to as postbiotics. In a B16-F0 preclinical melanoma model, the probiotic *Lactobacillus reuteri*, translocated from the gut to the tumor, colonizing it. Within the tumor mass, *L. reuteri* released indole-3-aldehyde (I3A), a dietary tryptophan catabolite, which promoted interferon gamma (IFN-γ) production through activation of the AhR within CD8+ T cells, thus, enhancing anti-tumoral immunity and boosting anti-PD-L1 efficacy (Bender et al, 2023). Similarly, a *Lactobacillus paracasei*-derived postbiotic upregulated major histocompatibility complex (MHC) class-I expression on cancer cells, increased antigen-specific T-cell

## A. Anti-tumoral effect

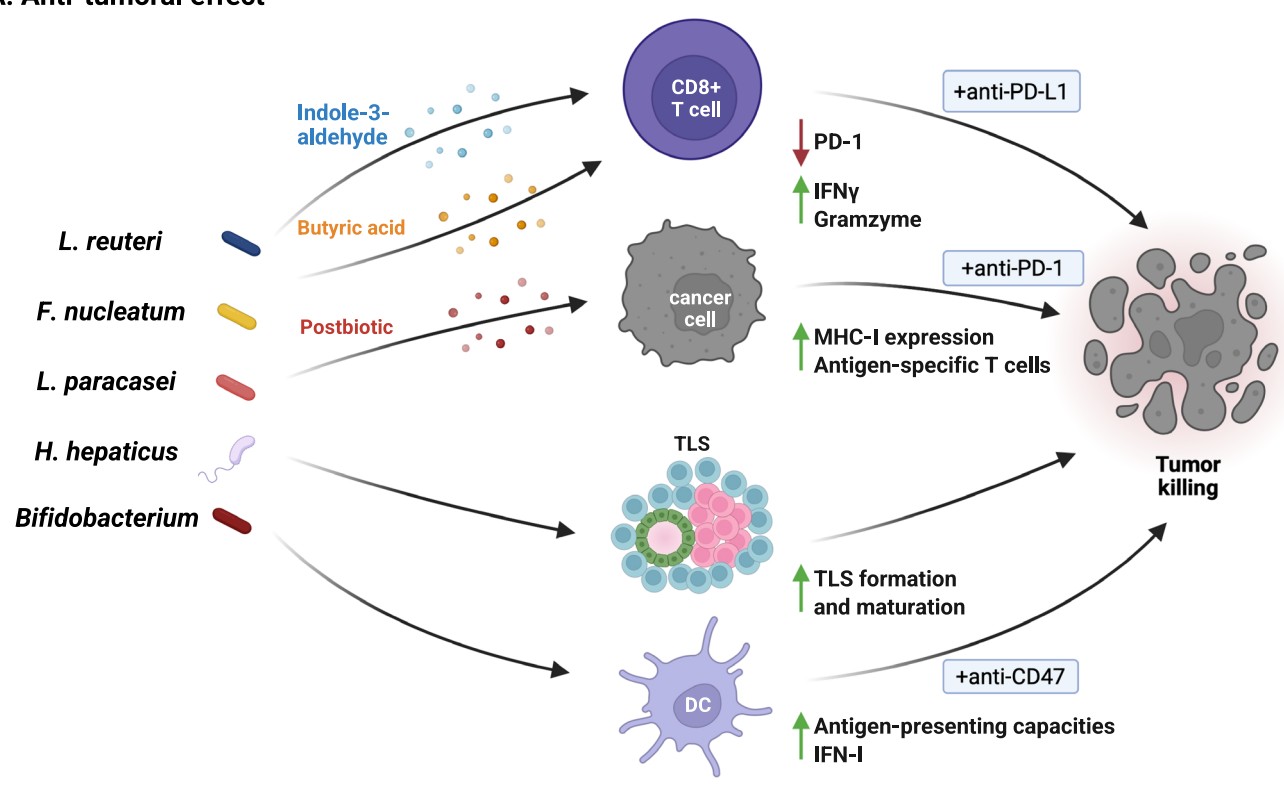

## B. Pro-tumoral effect

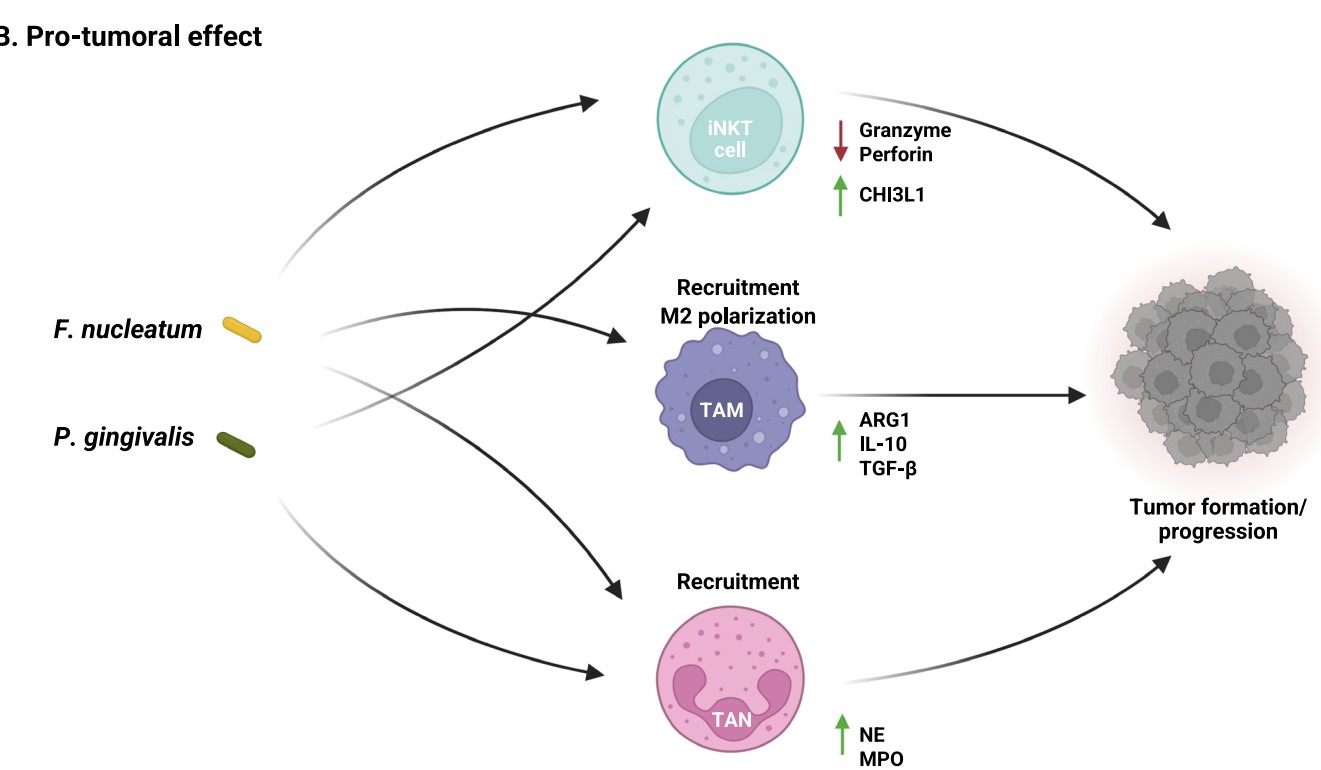

◄ **Figure 3. Anti- and pro-tumoral immunity regulated by tumor-resident bacteria.**

(**A**) Bacteria residing in tumors can promote anti-tumoral immunity through various mechanisms. (1) CD8+ T-cell activation: *Lactobacillus reuteri* and *Fusobacterium nucleatum* enhance CD8+ T-cell activity by releasing the metabolic products (postbiotics) indole-3-aldehyde (I3A) and butyric acid, respectively. I3A activates the aryl hydrocarbon receptor (AhR) in CD8+ T cells, stimulating interferon gamma (IFN-γ) production and improving anti-PD-L1 therapy efficacy. Butyric acid, on the other hand, triggers a signaling pathway involving TBX21 (T-bet) in CD8+ T cells, resulting in PD-1 suppression and increased T-cell effector activity. (2) Modulation of major histocompatibility complex (MHC) class-I expression: a postbiotic derived from *Lactobacillus paracasei* elevates MHC class-I expression on cancer cells, promoting the recruitment and activation of antigen-specific T cells, ultimately enhancing anti-PD-1 immunotherapy efficacy. (3) Tertiary lymphoid structures (TLS) induction: *Helicobacter hepaticus* drives the formation and maturation of tumor-associated TLSs enhancing anti-tumoral immunity through CD4+ T follicular helper cells and B cells, leading to tumor growth control. (4) Improvement of antigen-presenting capacities: *Bifidobacterium* boosts antigen-presenting capabilities of dendritic cells (DC) through STING signaling, promoting a favorable response to anti-CD47 immunotherapy. (**B**) Tumor-resident bacteria can favor pro-tumoral immunity in several ways. (1) iNKT conversion to a pro-tumoral phenotype: *Porphyromonas gingivalis* and *F. nucleatum* modulate iNKT cell activity. *P gingivalis* hampers iNKT cytotoxicity by upregulating the chitinase 3-like-1 protein (CHI3L1), leading to reduced expression of cytotoxic cytokines. (2) Macrophages recruitment: *F. nucleatum* favors infiltration of macrophages via CCL20 and their polarization to the pro-tumoral M2 phenotype. (3) Recruitment of tumor-associated neutrophils (TAN): *P. gingivalis* and *F.nucleatum* drive the accumulation of TANs within the TME. *P. gingivalis* induces the upregulation of neutrophil-attracting chemokines and the release of neutrophil elastase (NE) by TANs, promoting cancer development.

activation and trafficking to the tumor, eventually boosting anti-PD-1 immunotherapy efficacy in a 4T1 breast cancer mouse model (Ferrari et al, 2023). Importantly, bacteria can exert divergent and context-dependent effects on anticancer treatment outcomes, underscoring the complexity of host-microbiome interactions in cancer therapy. While *F. nucleatum* has been well-documented as a contributor to chemoresistance in CRC (Yu et al, 2017; Zhang et al, 2019), recent findings reveal a paradoxical role for this bacterium in enhancing the efficacy of immunotherapies. Specifically, *F. nucleatum* has been shown to improve the response to anti-PD-1 therapy in microsatellite-stable (MSS) CRC, a subtype that is typically resistant to immune checkpoint inhibitors. Mechanistically, intratumoral *F. nucleatum* produces butyric acid, which initiates a signaling cascade mediated by TBX21 (T-bet) in CD8+ tumor-infiltrating lymphocytes (TILs), ultimately leading to PD-1 repression and enhanced T-cell effector function (Wang et al, 2024). Thus, the same bacterium can have opposing effects on cancer treatment depending on the therapeutic context; while *F. nucleatum* promotes resistance to chemotherapy, it can enhance immunotherapy efficacy. Another bacterium reported to enhance immunotherapy efficacy in CRC is *Bifidobacterium*. At the tumor site, *Bifidobacterium* improves antigen-presenting capacities in dendritic cells (DC) via STING signaling, favoring anti-CD47 immunotherapy response, in a type I IFN-dependent manner (Shi et al, 2020).

TRB can also induce tumor-associated tertiary lymphoid structures (TLS) formation. TLSs are ectopic lymphoid aggregates occurring upon chronic inflammation and generally associated with a positive prognosis for many tumor types, including CRC (Di Caro et al, 2014; Petitprez et al, 2020; Sautes-Fridman et al, 2019). Colonization of an immunogenic bacterium, *Helicobacter hepaticus*, has been shown to drive the induction and maturation of TLSs surrounding and within CRC in a carcinogen-induced orthotopic mouse model. Moreover, colonization of *H. hepaticus* favored anti-tumoral immunity mediated by CD4+ T follicular helper cells and B cells, ultimately resulting in tumor growth control (Overacre-Delgoffe et al, 2021).

Another bacterial-mediated mechanism of stimulating anti-tumoral response involves the presentation of bacterial peptides on the surface of cancer cells via the Human Leukocyte Antigens (HLA) class-I and -II molecules (known as MHC-I and -II in mice). These molecules are essential for immune recognition. HLA class-I presents intracellular antigens to CD8+ cytotoxic T cells, enabling direct killing of target cells presenting these non-self antigens. HLA class-II presents extracellular or phagocytosed bacterial antigens to CD4+ helper T cells, which are essential for orchestrating broader immune responses, including the activation of CD8+ T cells and the production of pro-inflammatory cytokines (Kotsias et al, 2019). The presentation of bacterial peptides via these molecules allows the immune system to recognize and potentially eliminate cancer cells harboring intracellular bacteria. Two pioneering studies in melanoma and in glioblastoma (GBM) underscore the potential of TRB to contribute to immune responses through this mechanism. The work led by Kalaora and Nagler was probably the first to identify a peptide repertoire derived from intracellular bacteria presented on HLA-I and HLA-II in tumors. Through 16S rRNA gene sequencing coupled to HLA peptidome analysis of 17 melanoma metastases, revealed 248 and 35 unique HLA-1 and HLA-II peptides, respectively, derived from 41 distinct bacterial species. In vitro assays demonstrated that certain bacterial peptides were immunogenic, as they increased IFN-γ secretion by tumor-infiltrating lymphocytes (TILs) when loaded into Epstein-Barr virus-transformed B cells, thereby eliciting immune reactivity (Kalaora et al, 2021). Similarly, 16S rRNA gene sequencing and HLA peptidome analysis in 19 primary and recurrent glioblastoma samples identified a total of 37 HLA-II-bound bacterial peptides, with four shared peptides between primary and recurrent tumors. Three bacterial peptides were shown to stimulate T-cell reactivity, inducing the secretion of pro-inflammatory cytokines such as TNF and IFN-γ by TILs (Naghavian et al, 2023). Given that bacterial peptides represent non-self-antigens, their ability to provoke T-cell responses suggests they could serve as novel targets for immunotherapy.

## Tumor-resident bacteria are implicated in the metastatic process

The metastatic establishment of cancer cells at distant sites is a complex and multistep process requiring cancer cells to acquire motility, intravasate into the bloodstream—either directly or via the lymphatic system—extravasate, proliferate, and ultimately colonize secondary organs (Vanharanta and Massagué, 2013). These steps in the metastatic cascade are governed not only by the intrinsic properties of the cancer cells themselves but also by interactions with microorganisms present within the TME.

Emerging evidence highlights the significant role of TRB in the metastatic cascade. Bullman et al. reported the persistence of specific bacteria, including *F. nucleatum, B. fragilis*, and *Prevotella* species, in both primary colorectal cancer (CRC) tumors and their corresponding liver metastases. Remarkably, *Fusobacterium* species isolated from matched primary and metastatic tumor pairs were found to be viable and genetically identical, despite the two tissues being collected months or even years apart (Bullman et al, 2017). A similar phenomenon was observed in a murine model of breast cancer, where *Staphylococcus xylosus* was detected in 80% of metastatic lung tissues, but was absent in non-metastatic lungs (Lu et al, 2022). These observations suggest that certain TRB persist during metastasis, migrating from primary tumors to metastatic sites, and may actively contribute to the metastatic process (Bullman et al, 2017).

Although the precise roles of these bacteria during metastasis remain to be fully elucidated, accumulating data indicate that they are not merely passive 'passengers' but active participants in regulating the metastatic process.

One proposed mechanism by which bacteria influence metastasis involves the modulation of host cell adhesion and the epithelial-mesenchymal transition (EMT) (Galeano Niño et al, 2022; Lu et al, 2022; Zhang et al, 2022). For example, *F. nucleatum* has been shown to facilitate cancer cell adhesion to vascular endothelial cells and promote extravasation by upregulating intercellular adhesion molecule 1 (ICAM-1) on CRC cells via activation of the NF-κB signaling pathway (Zhang et al, 2022). Furthermore, *F. nucleatum* enhances the migratory capabilities of CRC cells by activating signaling pathways that govern cell adhesion, migration, extracellular matrix remodeling, and EMT, as demonstrated in both in vitro and in vivo CRC models (Galeano Niño et al, 2022; Lu et al, 2022). This reprogramming is linked to the upregulation of long non-coding RNA EVADR, which scaffolds Y-box binding protein 1 (YBX1) to enhance the translation of EMT-associated factors in host cells (Lu et al, 2022).

Another mechanism involves the reorganization of the host cell cytoskeleton (Fu et al, 2022; Oliveira et al, 2003). In a spontaneous PyMT murine breast cancer model, intracellular bacteria within circulating cancer cells promote survival and metastatic colonization in the lungs by counteracting fluid shear stress through actin cytoskeleton reorganization (Fu et al, 2022). Similarly, a soluble factor identified as a 13mer beta-casein-derived peptide released by *Listeria monocytogenes* triggers actin rearrangement and filopodia formation, thereby enhancing the motility of HCT-8/E11 CRC cells (Oliveira et al, 2003).

Finally, intratumoral bacteria can facilitate the establishment of a pre-metastatic niche. CRC mouse models revealed that tumor-resident *Escherichia coli C17* disrupts the gut vascular barrier (GVB) and translocates to the liver. In the liver, these bacteria promoted the formation of an inflammatory environment conducive to a pre-metastatic niche, thus facilitating cancer cell seeding and metastasis establishment. This finding was further corroborated in CRC patients, where increased GVB permeability, assessed by PV-1 expression levels, was associated with metachronous distant metastases, independent of lymph node involvement (Bertocchi et al, 2021). These observations suggest that the lymphatic and vascular systems represent distinct routes for cancer cell dissemination, with bacteria potentially migrating ahead of tumor cells to create a pre-metastatic niche that supports the subsequent cancer cell seeding. Notably, bacteria may also migrate within cancer cells to metastatic sites (Fu et al, 2022; Galeano Niño et al, 2022).

Metastatic lesions do not occur randomly but rather exhibit organ-specific patterns. This phenomenon was first described in Paget's "seed and soil" hypothesis in the late 19th century, which posits that metastases arise from favorable interactions between metastatic cancer cells (the "seed") and the organ microenvironment (the "soil") (Paget, 1989). For instance, breast and prostate cancer preferentially metastasize to the bone, colon cancer to the liver, while ovarian cancer to the peritoneum (Bubendorf et al, 2000; Budczies et al, 2015; Gao et al, 2019; Lee, 1985; Riihimäki et al, 2016; Rose et al, 1989). More recently, microbiome analyses of over 4000 pre-treatment tumor biopsies, conducted through whole-genome sequencing (WGS), have revealed that the microbial community composition is more strongly influenced by the anatomical site of the biopsied lesion than by the primary tumor type (Battaglia et al, 2024). This suggests the presence of organ-specific microbial tropisms, which may further influence the metastatic process. For instance, certain bacteria might preferentially colonize the liver or bone due to tissue-specific nutrient availability, immune permissiveness, or chemotactic cues, thereby creating a microenvironment that is more favorable for metastatic seeding. While organ-specific tropism represents an emerging area of investigation within the study of tumor-associated microbiomes, further research is essential to fully elucidate its implications. Comprehensive comparisons of the microbial communities in primary tumors and their corresponding metastases could lead to the identification of key bacterial species that may serve as biomarkers for predicting the development of distant metastatic lesions.

## Exploiting bacteria for therapeutic intervention

Several innovative strategies now leverage microbes as tools for therapeutic intervention in cancer and other diseases (Fig. 4). Among these, fecal microbiota transplantation (FMT) and the administration of engineered probiotic bacteria have emerged as promising approaches. FMT, the transfer of fecal bacteria from a healthy donor to a recipient to modulate gut microbiota, has shown promising results in preclinical and clinical cancer settings (Baruch et al, 2021; Davar et al, 2021; Gopalakrishnan et al, 2018; Routy et al, 2018; Wang et al, 2024). However, possible changes in TRB communities upon FMT need to further investigation. Probiotic therapy, particularly through genetically engineered bacteria, has taken advantage of synthetic biology's rapid advances. This field has enabled the reprogramming of live bacteria into safe, immune-stimulating carriers for delivering therapeutic agents—including cytokines, nanobodies, and metabolic products—directly to tumor sites, thereby minimizing off-target effects (Canale et al, 2021; Chowdhury et al, 2019; Leventhal et al, 2020; Li et al, 2024; Savage et al, 2023; Zhu et al, 2022). A notable example is the intratumoral delivery of an engineered *E. coli* Nissle 1917 strain designed to produce interferon gamma (IFN-γ), which has shown efficacy in a murine model of MC38 colorectal cancer. This approach leverages a genomically integrated synchronized lysis circuit (SLIC–IFN-γ) that enables quorum-regulated release of IFN-γ specifically within

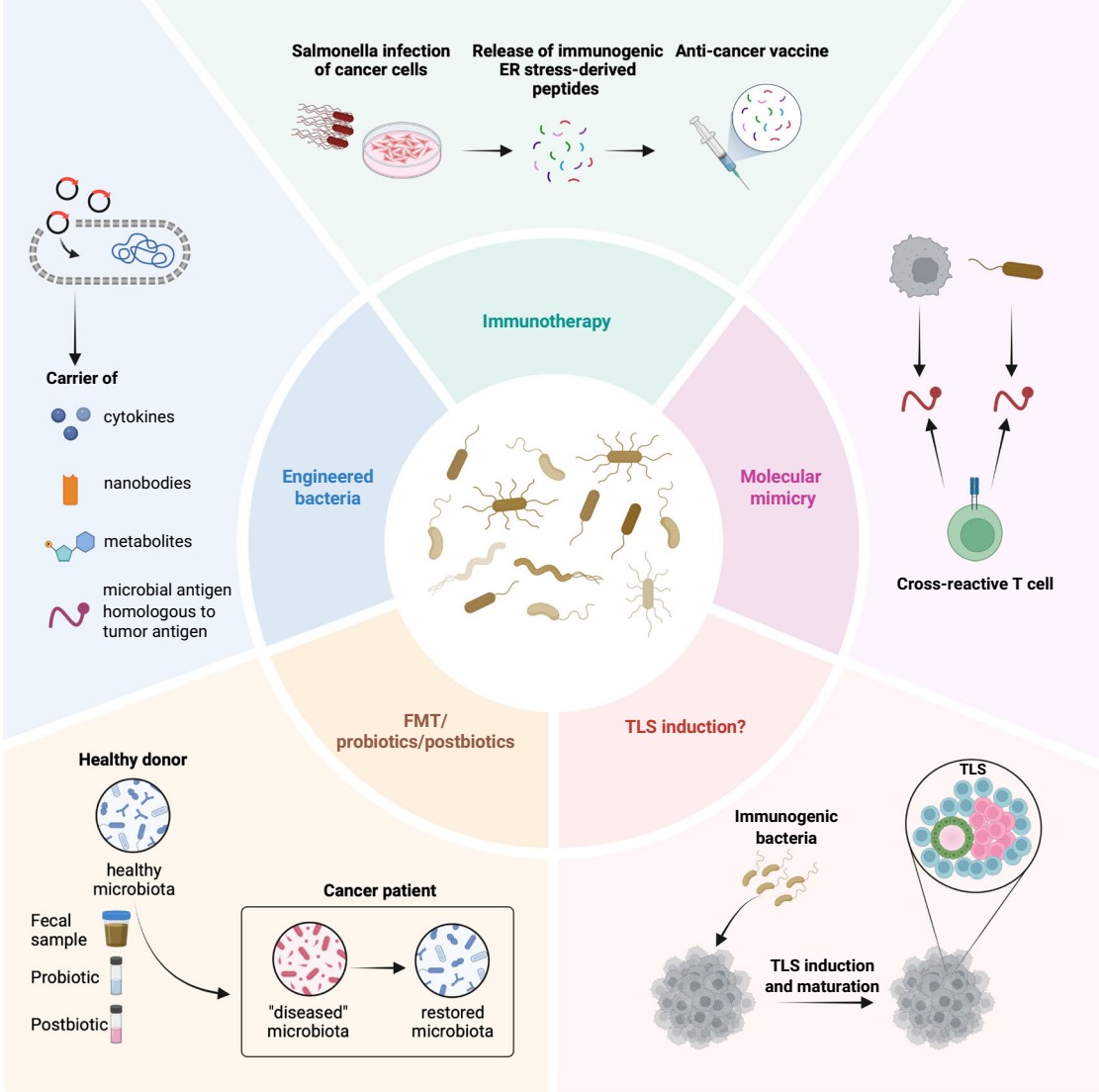

**Figure 4. Utilizing bacteria for therapeutic intervention in cancer.**

Various approaches utilize bacteria as instruments for therapeutic cancer treatment. (1) Bacteria can be engineered to deliver therapeutic agents such as cytokines, nanobodies, and metabolic products directly to tumor sites. Additionally, engineered bacteria can express antigens that structurally resemble tumor antigens. (2) Bacteria can be exploited for developing immunotherapy approaches. Infection of cancer cells with an avirulent strain of *Salmonella* triggers the release of immunogenic peptides associated with endoplasmic reticulum (ER) stress response, offering a promising new source for peptide vaccine development. (3) Certain bacteria intrinsically exhibit molecular mimicry, enabling the generation of T cells that recognize and target both microbial and tumor antigens, potentially enhancing antitumor immunity. (4) Administration of immunogenic bacteria, such as *Helicobacter hepaticus*, could induce the formation and maturation of tumor-associated tertiary lymphoid structures (TLS). (5) Fecal microbiota transplantation (FMT), which involves the transfer of fecal bacteria from a healthy donor to a recipient, as well as the administration of probiotics or postbiotics, can be used to modulate the gut microbiota. Although it has shown promising outcomes in preclinical and clinical cancer settings, the alterations in tumor-resident bacterial communities following FMT, probiotics, or postbiotics require further investigation.

the tumor environment. The engineered bacteria accumulate in the tumor, where quorum-induced lysis is triggered, releasing IFN-γ locally and thereby minimizing systemic toxicity. This strategy underscores the potential of bacteria as precise, targeted vehicles for therapeutic agent delivery (Li et al, 2024).

Bacteria can also be engineered to express peculiar microbiota-derived antigens that share significant homology with tumor antigen, a phenomenon called "molecular mimicry". This shared structural resemblance allows cross-reactive T cells to recognize

and respond to both microbial and tumor antigens, a promising avenue for enhancing antitumor immunity (Chiaro et al, 2021; Fluckiger et al, 2020; Rubio-Godoy et al, 2002; Tagliamonte et al, 2023). For instance, a molecular mimicry has been demonstrated between the tape measure protein (TMP) epitope of a prophage and the oncogenic PSMB4 epitope of murine tumors. Expression of the TMP epitope in an engineered *Enterococcus hirae* was successful to improve response to the chemotherapeutic agent cyclophospha-mide and the PD-1 blockade immunotherapy, stimulating T-cell

responses against TMP and PSMB4 antigens, in mouse models of MCA205 sarcoma and TC1 lung tumors (Fluckiger et al, 2020). Molecular mimicry has also been observed in a recent study, where three flagellin-related Lachnospinaceae (FLach) peptides showing structural homology with the melanoma-associated antigens MART1/Melan-A, SCRN1, and PRAME improved anti-tumoral immune response against patient-derived melanoma organoids (Macandog et al, 2024). Thus, leveraging molecular mimicry to prime the immune system against tumors might offers an additional bacterial-based strategy in cancer.

Moreover, in vitro studies reveal that infection of melanoma cells with an avirulent strain of *Salmonella* can induce the endoplasmic reticulum (ER) stress response, resulting in the release of melanoma-specific immunogenic peptides into the extracellular environment. This newly characterized class of immunogenic peptides, termed ER stress-derived peptides, represents a novel source for vaccine development that could train T cells to specifically recognize and kill melanoma cells. Although *Salmonella* is not traditionally classified as a TRB, this work suggests an unexplored strategy for using bacteria to develop novel immunotherapy approaches (Melacarne et al, 2021).

Finally, given that the development of tumor-associated TLSs may be influenced by the immunostimulatory properties and overall immunogenicity of the TME (Peske et al, 2015), introduction of immunogenic bacteria, such as *H. hepaticus*, could help to induce formation and maturation of these structures, thereby boosting antitumor immune responses and potentially improving outcomes in immunotherapy (Overacre-Delgoffe et al, 2021).

## Challenges and advancements in profiling tumor-resident bacteria

NGS technologies, such as whole-metagenome shotgun sequencing (WMS) and 16S rRNA gene sequencing, have emerged as powerful tools for deeply characterizing microbial communities. WMS analyzes the entire microbial genome, while 16S rRNA gene sequencing focuses on the amplification of one or multiple regions of the 16S rRNA gene, which is specific to prokaryotes (Pérez-Cobas et al, 2020; Sanschagrin and Yergeau, 2014). These technologies are particularly effective for profiling bacterial communities in samples with minimal host DNA contamination, such as stool samples, while their application to tumor samples presents significant challenges. This is due to very low levels of bacterial DNA in tumors-several orders of magnitude less than in the gut microbiome-and the overwhelming presence of host genomic DNA, making it difficult to distinguish bacteria genuinely present in the tumor from those introduced through experimental contamination during sample collection and processing (Dohlman et al, 2021; Eisenhofer et al, 2019). To address these challenges, careful experimental design is critical. Incorporating appropriate controls, such as environmental and extraction controls, helps to identify and account for potential contaminants introduced during the workflow. Similarly, rigorous sterile DNA extraction procedures are essential to minimize contamination and improve the reliability of bacterial profiling in tumor samples. The prokaryotic 16S rRNA gene is ~1500 bp long, with nine variable regions (V1–V9) that are interspersed with conserved regions. To improve both coverage and resolution in bacterial species detection, sequencing multiple

hypervariable regions of the 16S rRNA gene, instead of the widely used V4 or V3–V4 sequencing, is necessary (Hilmi et al, 2023; Nejman et al, 2020). Nejman et al. developed the 5R 16S rDNA sequencing method, which amplifies five short regions of the 16S rRNA gene and enables the amplification of 68% of the bacterial 16S rRNA gene. This method successfully facilitated the bacterial characterization of seven cancer types using formalin-fixed paraffin-embedded (FFPE) samples (Nejman et al, 2020). WMS offers higher resolution than 16S rRNA sequencing, providing accurate insights into microbiota diversity and their biological functions. However, 16S rRNA gene sequencing remains the most widely used approach for studying TRB due to its relatively low cost and good accessibility (Lu et al, 2024; Ranjan et al, 2016). Moreover, on the contrary of the 16S rRNA gene sequencing, performing WMS in tumor samples requires microbial enrichment methods to reduce host DNA contamination. Different kits of microbial DNA enrichment have been developed and are commercially available employing either chemical lysis of host DNA or physical removal of host DNA using magnetic beads (Marchukov et al, 2023). Identifying the optimal microbial enrichment method that effectively enriches bacterial DNA with minimal or no alterations to the microbial community will be crucial for performing WMS on tumor samples and obtaining metatranscriptomic information from TRB. Alongside with NGS, imaging-based approaches, such as fluorescence in situ hybridization (FISH), immunohistochemistry, and electron microscopy are commonly used to assess TRB presence and abundance in cancer specimens (Fu et al, 2022; Galeano Niño et al, 2022; Lu et al, 2024; Nejman et al, 2020). It is worth mentioning that detecting bacterial genomic material through FISH and NGS does not provide definitive evidence of bacterial viability. Culturomics, which consists in the isolation and identification of viable tissue-resident bacteria by applying specific culture conditions, represents a promising complementary approach. An additional challenge arises from the fact that the above-mentioned technologies are predominantly focused on identifying correlations between TRB and certain cancer features, rather than establishing causality. To address this limitation, both in vitro and in vivo experiments are essential. An important aspect of this research is improving co-culture protocols for bacteria with cancer cells (2D system) or with spheroids and organoids (3D system), as these advancements enable the detection of microbial-host interactions. In gastrointestinal research, several approaches involving organoids are commonly used (Puschhof et al, 2021). One approach is based on the microinjection of bacteria into the lumen of intestinal organoids (Williamson et al, 2018). This method is particularly useful for maintaining the viability of anaerobic bacteria and ensuring direct contact between the microbes and the apical side of the gut epithelium, thereby mimicking their natural interaction site with host cells. However, this technique is technically challenging and requires specialized and expensive equipment. A more recently described technique involves the use of "inverted" organoids, in which the apical surface faces outward, resulting in reversed polarity. In this configuration, microbes can be simply added to the culture medium to interact directly with the organoid's apical side (Co et al, 2019). This approach facilitates the analysis of secretions (e.g., mucus) from the apical surface of the epithelium. Nevertheless, a major limitation of this method is the slower proliferation and accelerated differentiation of the organoids, which leads to a

**Table 3. Advantages and limitations of current methods for analyzing TRB.**

| Method | Advantages | Limitations |
|---|---|---|
| Metagenomics | • High resolution<br>• Provides functional insights | • Expensive<br>• Not applicable to samples contaminated by host DNA<br>• No definitive evidence of bacterial viability |
| 16S rRNA gene sequencing | • Applicable to samples contaminated by host DNA<br>• Low cost | • Limited resolution<br>• Risk of false positive in low-biomass samples<br>• Need of appropriate controls<br>• Need of rigorous sterile DNA extraction procedure<br>• No definitive evidence of bacterial viability |
| Imaging (FISH, electron microscopy) | • Maintained spatial organization | • No definitive evidence of bacterial viability |
| Culturomics | • Viable microbial isolates | • Influenced by culture media and conditions (e.g., $O_2$ levels)<br>• Labor-intensive |
| 2D co-culture | • Inexpensive<br>• Easy to use | • Loss of intratumoral spatial context<br>• Limited to (facultative) aerobic bacteria due to oxygen requirements |
| 3D co-culture (organoids) | • Reflects patient-specific background<br>• Enables study of microbial-host interactions<br>• Applicable to aerobes and anaerobes (microinjection)<br>• Allows analysis of secretions (inverted) | • Time-consuming<br>• Technically challenging (microinjection)<br>• High equipment cost (microinjection)<br>• Limited experimental time frame (inverted) |
| Mouse models | • Host–pathogen interactions<br>• Useful for studying tumor growth and progression | • Time-consuming |

reduced experimental time window. In vivo studies, on the other hand, allow to observe dynamic interactions within a more complex biological system, providing insights into how TRB impacts tumor progression, metastasis, and treatment responses.

Overall, the techniques currently used to study TRB offer certain advantages but also have notable limitations (Table 3). To gain a more definitive understanding of the functional roles of TRB and their interactions within the TME, future methodological advancements will be essential. Given the extremely low microbial biomass in tumor samples, improved microbial DNA enrichment techniques and more sensitive sequencing methods will enable researchers to accurately profile the tumor-associated microbiota, minimizing host DNA interference. A combination of high-resolution sequencing, culturomics, and advanced imaging will provide a more comprehensive and descriptive view of the bacterial communities within tumors. These descriptive approaches will be instrumental in selecting specific bacterial candidates for use in co-culture systems and in vivo models, which are crucial for gaining mechanistic insights. Therefore, an integrative, multimodal strategy will be key to unraveling the complex roles of TRB in influencing tumor biology.

## Concluding remarks

TRB are components of the TME and regulate several aspects of the tumor biology. While certain TRB populations have beneficial roles, other ones tend to exert detrimental effects by reducing the efficacy of antitumor therapies and promoting tumor development and metastasis, underscoring their dual role in cancer. Therapeutic strategies aimed at depleting detrimental TRB, such as the administration of antibiotics or bacteriophages, could potentially be effective. However, the use of antibiotics in this context must be carefully considered, as their broad-spectrum nature poses the risk

of eliminating the beneficial TRB as well as inducing dysbiosis due to their lack of specificity in targeting both particular bacterial species and specific organs. In preclinical models, a promising strategy of antibiotic-specific targeting of tumor resident bacteria is represented by liposomes loaded with a silver-tinidazole complex (LipoAgTNZ) in that its administration eliminated anaerobe microbes in primary CRC tumor and liver metastases without causing gut microbiome dysbiosis (Wang et al, 2023). However, further investigation is needed in cancer patients. Bacteriophages, or phages, are viruses that specifically infect and kill bacteria, offering a promising alternative to antibiotics. Unlike antibiotics, bacteriophages can penetrate the dense, protective matrix of bacterial biofilms, making them particularly interesting for targeting biofilm-associated bacteria, such as *F. nucelatum* (Kabwe et al, 2019; Kabwe et al, 2021). Although bacteriophage therapy may offer significant advantages, challenges related to bio-distribution, limited penetration in solid tumors, and clearance by the immune system remain to be addressed.

Despite the significant technical advancements made in recent years in studying TRB, much of their role in cancer biology remains unexplored (see Box 1). Key questions persist regarding the causal, temporal, and spatial relationships between TRB, other components of the TME, and their downstream effects on tumor development, progression, and therapeutic responses. Addressing these complex interactions requires a multimodal methodology, consisting of imaging-based techniques, sequencing, culturomics as well as in vivo and in vitro studies. Finally, possible microbe-microbe interactions within the TME, including quorum sensing and competitive or synergistic bacterial interplays, that can influence cancer biology still remain unexamined. Similarly, the role of TRB within immune cells, including their potential impact on immune modulation and response to immunotherapy, also requires further investigation. Unraveling the complexities of TRB interactions and functions within the TME could lead to the

**Box 1   In need of answers**

- Can the source of TRB be determined for each cancer type?

- What evolutionary pressures drive the adaptation of certain bacteria to the tumor niche?

- What are the causal, temporal, and spatial relationships between TRB and other components of the TME, such as immune cells, stromal cells, and tumor vasculature?

- Do certain bacteria exhibit tropism for specific metastatic sites, and what factors govern their site-specific colonization?

- Considering the dual role of bacteria in cancer, can we selectively eliminate detrimental TRB—e.g., via phage therapy or antibiotics—while preserving beneficial or commensal bacteria?

- Can intratumoral microbial signatures serve as reliable biomarkers for cancer prognosis and/or treatment response?

identification of new molecules or pathways that could be targeted to enhance antitumor responses as well as the development of innovative bacteria-based therapies.

# Peer review information

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

## Acknowledgements

This work was supported by the European Molecular Biology Organization (GV; ALTF 687-2023) and the Fondazione AIRC per la ricerca sul cancro ETS (MR; AIRC 5×1000 22757 and AIRC IG 28930-2023). Figures were created with Biorender.com.

## Author contributions

**Gerlanda Vella**: Conceptualization; Writing—original draft; Writing—review and editing. **Maria Rescigno**: Conceptualization; Writing—original draft; Writing—review and editing.

## Disclosure and competing interests statement

MR is the founder and CSO of Postbiotica srl.

