## [Peer Review File · EMBO Reports]

Cancer microbiota: a focus on tumor-resident bacteria

Gerlanda Vella and Maria Rescigno

Corresponding author(s): Maria Rescigno (maria.rescigno@hunimed.eu)

Review Timeline:

Submission Date:	31st Jan 25
Editorial Decision:	25th Mar 25
Revision Received:	28th Apr 25
Accepted:	13th May 25

Editor: Achim Breiling

Transaction Report:

Dear Prof. Rescigno,

Thank you for the submission of your review article to our editorial offices. I have now received the full set of referee reports that is copied below. As you will see, all three referees state that your manuscript is interesting and timely. However, they have several suggestions to improve the submission that I kindly ask you to address in a revised manuscript.

Given the constructive referee comments, I would thus like to invite you to revise your manuscript with the understanding that all referee points will be addressed in the revised manuscript and in a detailed point-by-point response.

I further have these editorial requests:

- Please add up to 5 keywords to the manuscript and place these below the abstract.
- Please mark the corresponding author on the title page and provide an e-mail contact.
- We updated our journal's competing interests policy in January 2022 and request authors to consider both actual and perceived competing interests. Please review the policy <https://www.embopress.org/competing-interests> and update your competing interests if necessary. Please name this section 'Disclosure and Competing Interests Statement'.
- We usually ask our authors to include a box called "In need of answers" that briefly outlines the major questions that are still open in a given field in the form of a few bullet points. These questions can be accompanied by a brief explanation of what would be needed to address them and may provide helpful towards setting the stage for future experimentation in the field. For an example see this recent review we published: <https://www.embopress.org/doi/full/10.1038/s44319-024-00135-4>
- Please also add callouts for the box to the manuscript text (Box 1).
- Please make sure that all figure panels are called out separately and sequentially. Presently, there are no separate callouts for the panels of Figures 2 and 3. Please check.
- Please make sure that all the funding information is also entered into the online submission system and that it is complete and similar to the one in the acknowledgement section of the manuscript text file.

I think this is a very interesting review and while I appreciate that incorporating the referees' suggestions will still require some work, I am convinced that the article is worth it and will benefit from it.

When submitting your revised manuscript, we will require a Microsoft Word file (.doc) of the revised manuscript text including detailed figure legends (placed after the references), but without the figures.

Please provide the final figures as separate, high-resolution files (without their legends) as .pdf, .eps, .tif, or .jpg (one file per figure). Please finalize the drafts provided and make sure they accurately illustrate the key scientific concepts that you wish to show.

Please also note the following points:

- If there are certain aspects of your figure draft that are based upon assumptions or where the scientific data remains ambiguous (for example, schematically depicting a presumed direct protein-protein interaction, protein shape or subcellular localizations etc.) please add a comment so that we can work with you on an accurate depiction. Please ensure the directionality and nature of interactions is presented accurately.
- If the figure or single panels of the figure have been adapted from a published figure, please add this information to the figure legend (e.g., 'Adapted from...' or 'Based on...'). The editor will discuss if a reference and permission will be necessary
- Please only re-use figures or parts of a figure if this is essential for understanding the concept communicated. Often a reference to a previous paper will suffice. If the figure contains re-used images or elements of images, including schematics, micrographs or photos, please make sure that you have the permission/license to publish it (this also applies to your own previous work, if the journal you published in retains copyright. Certain 'creative commons' open access licenses, such as CC-BY 4.0, allow re-use without additional formal permissions). All re-used material must be explicitly cited.
- If you use an image data base for scientific iconography (e.g., BioRender), please let us know if you have a license that allows

for publication in an academic journal. Often authors use misleading iconography for expedience. Please ensure the information shown is scientifically accurate. If in doubt, please discuss with the editor or provide a sketch so that our designers can create accurate iconography. Please acknowledge the use of BioRender once in the Acknowledgements section (not in the figure legend).

- For figures created using a software for editing vector objects like Inkscape, CorelDraw etc., please send the file as a PDF (or SVG, or EPS), PowerPoint or Keynote in which the labels and objects are still editable. For figures created using Adobe Illustrator, please send the Illustrator (.ai) file.

I look forward to seeing a revised version of your manuscript when it is ready. Please let me know if you have questions or comments regarding the revision.

Kind regards,

Achim

Referee #1:

This manuscript reviews on the role of tumor resident bacteria in cancer progression, immune response, and therapeutic outcomes. This is an important topic and the article nicely summarizes current progress in the field. Some changes may further improve the manuscript and make the reader benefit even more:

1. Greater attention should be given to spelling and grammar, as there were many mistakes throughout the manuscript that detract from the data shared

2. The authors may consider revising the order of the sections - possibly moving the "origin of tumor-resident bacteria" to earlier in the manuscript, as this is a question that pops-up almost instantly when reading the manuscript and does not make much sense to leave until near the end of the review. Additionally, greater mechanistic insight could be shared in this section, as it is crucial for setting the precedent of the review. Moreover, it would be helpful for the authors to highlight the currently, or most widely, accepted view of the field.

3. the manuscript would be better suited to incorporate a few tables, which would concisely organize the information presented.

Tables summaries to incorporate:

- i. TRB regulation of tumor initiation and progression - bacteria/factor(s) influencing cancer/how it influences cancer and which pathways (pro- or anti-tumor)/signaling involved/references
- ii. TRB in cancer therapy - bacteria/therapy type/influence on therapy/references
- iii. The advantages and limitations of current methods for analyzing TRB

4. Consider adding key points to further address/expand upon in this review:

Key aspects of the TME that promote the retention and survival of bacteria. How bacteria can sediment in the high-pressure environment of the tumor and circumvent the immune response in both the blood (if this is indeed the route of translocation) and tumor.

Address greater mechanistic detail on pathways that bacteria use to perturb immune cell phenotype, function, and migration in the context of cancer initiation/progression and therapy.

Expand on current explanation/observations on organ-specific tropism of TRB.

Referee #2:

In their review, Vella & Rescigno discuss how tumor-resident bacteria (TRB) influence cancer progression through interactions with the tumor microenvironment, impacting immunity, therapy response, and metastasis. They cover key aspects such as TRB's role in carcinogenesis, effects on antitumoral drugs, immune modulation/cancer immunotherapy and metastatic spread. Additionally, they explore the potential origins of TRB and highlight challenges in the field. Overall, the review is well-written and provides a broad overview of critical questions in the field. However, some important literature is missing, and a few minor

revisions should be addressed. A subsection on how TRB research could contribute to cancer prevention would be a valuable addition.

1. Intro

The authors should define 16S rRNA as it appears for the first time in the review. While it is discussed in detail in the challenges section, adding a couple of sentences in the introduction would help non-expert readers.

2. TRB and Carcinogenesis

This section lacks depth and does not adequately cover key recent advancements in the field. Several important studies are missing, such as:

- *Fusobacterium nucleatum* (Fn) has been identified in sites beyond colorectal cancer, including breast cancer (Parhi et al., Nat Com, 2020, PMID: 32591509).
- Fn affects cancer stem cells via formate production (Ternes et al., 2022, PMID: 35437333).
- The modulation of aspirin by Fn and its implications for cancer prevention should be discussed (PMID: 33824205).

3. Metastatic Spread

The authors mention the pro-metastatic niche, do they mean pre-metastatic niche? If so, the terminology should be clarified. If they intend to differentiate the two, a discussion on the distinction would be helpful.

4. Origin of TRB

- The molecule GalGalNAc should be written in full upon first mention. Additionally, its presence in breast and other cancers should be discussed.
- The authors could also expand on how comparing oral, fecal, and tumor-resident bacterial strains' genomes might provide insights into TRB origins.

5. Challenges

- The discussion on 3D culture models should be expanded, addressing potential limitations and technical challenges, such as inverted organoids and injection methods.

Minor points:

- In the origin of TRB section, correct the typo: "without casing" → "without causing".
- All bacterial acronyms should follow the convention of introducing the full name first before using abbreviations.

Referee #3:

In this well-written review, the authors explore the complex role of the human microbiota, particularly focusing on tumor-associated bacteria (TRB) and their implications for cancer biology.

I think the authors can elaborate more on the mechanisms of Fn in CRC. For instance, it has been shown that Fn expresses amyloid FadA to stimulate CRC growth (Meng et al, EMBO Rep 2021). Another study reported that Fn preferentially binds Annexin A1-expression cells and further stimulates Annexin A1 expression via FadA (Rubinstein et al EMBO Rep 2019). Annexin A1 is upregulated in CRC, which explains why Fn is enriched in CRC. The authors mentioned 5-FU in the review. I want to point out that a previous study reported Annexin A1 expression in CRC caused resistance to 5-FU, consistent with Fn causing resistance to 5-FU (Onozawa et al, Onco Rep 2017).

As for origin of the bacteria, there have been several studies reporting Fn identified in CRC originate from the mouth, which is the natural habitat of Fn. Several studies showed that oral gavage of Fn induced tumor formation in Apc min mice (Kostic et al, Cell Host Micro 2013; Rubinstein et al, EMBO Rep 2019). Therefore, Fn may colonize in CRC via multiple pathways.

Additional comments are following:

Abstract Section:

1. In the phrase "tumor-promoting and tumor-suppressing pathways," consider adding a comma before "and" for clarity: "tumor-promoting, and tumor-suppressing pathways."
2. The phrase "requires the use of multi-modal technologies" could be clearer. Consider rephrasing to "requires the application of multi-modal technologies."
3. The abstract must include keywords.

Introduction Section:

1. In "the developments of next-generation sequencing (NGS) technologies, particularly the 16S rRNA gene sequencing," there should be a space between "the" and "16s."
2. In "the tumor-associated microbiota is now recognized as an integral part of the TME," consider changing "is now recognized" to "has now been recognized" for consistency with the past tense used throughout the article.
3. In "highly intricated and dynamic environment," "intricated" is not commonly used in this context; consider using "intricate" instead.

4. Some paragraphs, particularly in the Introduction, are quite long. Consider breaking them into shorter paragraphs for better readability.

Others:

1. Ensure consistent punctuation within citations. For example, there is an inconsistent use of punctuation before citations (such as missing commas) throughout the text. A common format is to place the citation at the end of the sentence before the period.
2. In the sentence** "CagA disrupts various intracellular signaling pathways involved in cell proliferation and apoptosis,"** correct "disrupts" to "disrupts."
3. "promotes of a pro-inflammatory microenvironment conducive to oncogenesis," remove "of." It should read: "promotes a pro-inflammatory microenvironment conducive to oncogenesis."
The phrase "enhancing cell proliferation and promoting promoted colonic tumorigenesis" has redundancy with the word "promoted." It should read: "enhancing cell proliferation and promoting colonic tumorigenesis."
4. The phrase "This activation occurs following bacterial binding and colonization of host cells via the bacterial surface protein TMPC, which interacts with the Annexin A2 (ANXA2) receptor on gastric epithelial cells." Consider rephrasing for conciseness: "This activation occurs after bacterial binding and colonization of host cells via the surface protein TMPC, which interacts with the Annexin A2 (ANXA2) receptor on gastric epithelial cells."
5. "In addition to modulating immune responses and signaling pathways, TRB can induce DNA damage," consider specifying "tumor-resident bacteria" (TRB) instead of just "TRB" once at the beginning for clarity.
6. The phrase "Overall, these observations indicate that bacteria contribute to cancer development through various mechanisms, playing a significant and multifaceted role in the malignant transformation of cells," can be more concise: "These observations indicate that bacteria significantly contribute to cancer development through various mechanisms, playing a multifaceted role in malignant transformation."
7. In the phrase: "Emerging evidences point into the direction," it should be "Emerging evidence points in the direction."
8. The sentence "While in vitro P. gingivalis-primed human iNKT cells displayed reduced expression of cytotoxic molecules such as granzyme B and perforin and impaired lytic degranulation," could be clearer if rephrased: "In vitro, P. gingivalis-primed human iNKT cells exhibited reduced expression of cytotoxic molecules, such as granzyme B and perforin, along with impaired lytic degranulation."
9. The phrase "this bacterium induces macrophage infiltration via CCL20-mediated pathways and promotes the polarization of macrophages toward an M2 phenotype, which is associated with a pro-tumoral role" could be streamlined: "This bacterium induces macrophage infiltration via CCL20-mediated pathways and promotes M2 polarization, associated with a pro-tumoral role."
10. When discussing "bacterial peptides on the surface of cancer cells via the Human Leukocyte Antigens (HLA) class-I and -II molecules," it might be beneficial to briefly explain the significance of this process for readers who may not be familiar with immunology.
11. In "Colonization of an immunogenic bacterium, Helicobacter hepaticus, have been shown to drive induction and maturation of TLSs," change "have" to "has." It should read: "Colonization of an immunogenic bacterium, Helicobacter hepaticus, has been shown to drive induction and maturation of TLSs."
12. The sentence "These observations suggest that certain TRB persist throughout the metastatic process, migrating from the primary tumor to metastatic sites, and may play a role in the cancer metastasis process." could be more concise: "These observations suggest that certain TRB persist during metastasis, migrating from primary tumors to metastatic sites, and may actively contribute to the metastatic process."
13. In the phrase "accumulate data indicate that they are not merely passive 'passengers' but active participants," change "accumulate" to "accumulating" for grammatical correctness. Also, consider simplifying to: "data indicate that they are not merely passive 'passengers' but active participants."
14. The phrase "this reprogramming is partly explained by the upregulation of the long non-coding RNA EVADR, which acts as a modular scaffold for Y-box binding protein 1 (YBX1), subsequently enhancing the translation of EMT-associated factors such as Snail, Slug, and Zeb in host cells" is a bit lengthy. Consider breaking it down for clarity or simplify by focusing on the key message: "This reprogramming is linked to the upregulation of long non-coding RNA EVADR, which scaffolds Y-box binding protein 1 (YBX1) to enhance the translation of EMT-associated factors in host cells."
15. The sentence "Studies on CRC mouse models revealed that tumor-resident Escherichia coli C17 can disrupt the gut vascular barrier (GVB) and translocate to the liver via the hematogenous route" could be streamlined: "CRC mouse models revealed that tumor-resident Escherichia coli C17 disrupts the gut vascular barrier (GVB) and translocates to the liver."
16. The sentence "This breach in the mucosal barrier is a potential source of TRB, particularly in tumors originating from mucosal sites with external cavity exposure, such as lung, pancreatic, colorectal, and cervical cancers" could be streamlined: "Compromised mucosal barriers may serve as a source of TRB, especially for tumors from mucosal sites with external cavity exposure, including lung, pancreatic, colorectal, and cervical cancers."
17. In "the administration of engineered probiotic bacteria have emerged as promising approaches", change "have" to "has"
18. The phrase "This is due to the very low levels of bacterial DNA typically contained in tumors-several orders of magnitude less than what is found in the gut microbiome-and the overwhelming presence of host genomic DNA, that makes it difficult to distinguish bacteria truly present in the tumor from those introduced through experimental contamination during sample collection and processing" can be streamlined: "This is due to very low levels of bacterial DNA in tumors-several orders of magnitude less than in the gut microbiome-and the overwhelming presence of host genomic DNA, making it difficult to distinguish bacteria genuinely present in the tumor from those introduced through experimental contamination during sample processing."

19. In "The prokaryotic 16S rRNA gene is approximately 1500 bp long, with nine variable regions (V1-V9) interspersed with conserved regions," the word "interspersed" can be made clearer by specifying "that are interspersed with conserved regions."
20. In the phrase "fluorescence in situ hybridization (FISH), immunohistochemistry, electron microscopy are commonly used to asses TRB presence and abundance," adjust to "assess" instead of "asses" and add a comma before "and" for correct punctuation: "fluorescence in situ hybridization (FISH), immunohistochemistry, and electron microscopy are commonly used to assess TRB presence and abundance."
21. Replace outdated references, such as those from 2000 or 1907, with more current ones.
22. At the end it would be interesting if you added this point to the manuscript, in light of the challenges detailed in your article regarding the low levels of bacterial DNA in tumor samples, how do you envision future methodological advancements-such as microbial enrichment techniques or sequencing technologies-contributing to a more definitive understanding of the functional roles of TRB in tumor biology and their interactions within the tumor.
23. The authors drew Fn as curved rod with polar flagellum, which is incorrect. Fn has no flagellum. It is a long filamentous rod with taped ends, i.e. like a fusiform (thus the name).

Dear Prof. Rescigno,

Thank you for the submission of your review article to our editorial offices. I have now received the full set of referee reports that is copied below. As you will see, all three referees state that your manuscript is interesting and timely. However, they have several suggestions to improve the submission that I kindly ask you to address in a revised manuscript.

Given the constructive referee comments, I would thus like to invite you to revise your manuscript with the understanding that all referee points will be addressed in the revised manuscript and in a detailed point-by-point response.

I further have these editorial requests:

- Please add up to 5 keywords to the manuscript and place these below the abstract.

We have added the following keywords: microbiota, intratumoral microbiota, tumor microenvironment, metastasis, therapy efficacy.

- Please mark the corresponding author on the title page and provide an e-mail contact.

The corresponding author has been marked and the e-mail contact has been provided.

- We updated our journal's competing interests policy in January 2022 and request authors to consider both actual and perceived competing interests. Please review the policy <https://www.embopress.org/competing-interests> and update your competing interests if necessary. Please name this section 'Disclosure and Competing Interests Statement'.

- We usually ask our authors to include a box called "In need of answers" that briefly outlines the major questions that are still open in a given field in the form of a few bullet points. These questions can be accompanied by a brief explanation of what would be needed to address them and may provide helpful towards setting the stage for future experimentation in the field. For an example see this recent review we published: <https://www.embopress.org/doi/full/10.1038/s44319-024-00135-4>.

Thank you for the suggestion. As requested, we have included a Box 1 titled "In need of answers" at the very end of the revised manuscript. This section outlines key open questions in the field.

- Please also add callouts for the box to the manuscript text (Box 1).

- Please make sure that all figure panels are called out separately and sequentially. Presently, there are no separate callouts for the panels of Figures 2 and 3. Please check.

- Please make sure that all the funding information is also entered into the online submission system and that it is complete and similar to the one in the acknowledgement section of the manuscript text file.

I think this is a very interesting review and while I appreciate that incorporating the referees' suggestions will still require some work, I am convinced that the article is worth it and will benefit

from it.

When submitting your revised manuscript, we will require a Microsoft Word file (.doc) of the revised manuscript text including detailed figure legends (placed after the references), but without the figures.

Please provide the final figures as separate, high-resolution files (without their legends) as .pdf, .eps, .tif, or .jpg (one file per figure). Please finalize the drafts provided and make sure they accurately illustrate the key scientific concepts that you wish to show.

Please also note the following points:

- If there are certain aspects of your figure draft that are based upon assumptions or where the scientific data remains ambiguous (for example, schematically depicting a presumed direct protein-protein interaction, protein shape or subcellular localizations etc.) please add a comment so that we can work with you on an accurate depiction. Please ensure the directionality and nature of interactions is presented accurately.
- If the figure or single panels of the figure have been adapted from a published figure, please add this information to the figure legend (e.g., 'Adapted from...' or 'Based on...'). The editor will discuss if a reference and permission will be necessary
- Please only re-use figures or parts of a figure if this is essential for understanding the concept communicated. Often a reference to a previous paper will suffice. If the figure contains re-used images or elements of images, including schematics, micrographs or photos, please make sure that you have the permission/license to publish it (this also applies to your own previous work, if the journal you published in retains copyright. Certain 'creative commons' open access licenses, such as CC-BY 4.0, allow re-use without additional formal permissions). All re-used material must be explicitly cited.
- If you use an image data base for scientific iconography (e.g., BioRender), please let us know if you have a license that allows for publication in an academic journal. Often authors use misleading iconography for expedience. Please ensure the information shown is scientifically accurate. If in doubt, please discuss with the editor or provide a sketch so that our designers can create accurate iconography. Please acknowledge the use of BioRender once in the Acknowledgements section (not in the figure legend).
- For figures created using a software for editing vector objects like Inkscape, CorelDraw etc., please send the file as a PDF (or SVG, or EPS), PowerPoint or Keynote in which the labels and objects are still editable. For figures created using Adobe Illustrator, please send the Illustrator (.ai) file.

I look forward to seeing a revised version of your manuscript when it is ready. Please let me know if you have questions or comments regarding the revision.

Kind regards,

Achim

Referee #1:

This manuscript reviews on the role of tumor resident bacteria in cancer progression, immune response, and therapeutic outcomes. This is an important topic and the article nicely summarizes current progress in the field. Some changes may further improve the manuscript and make the reader benefit even more:

1. Greater attention should be given to spelling and grammar, as there were many mistakes throughout the manuscript that detract from the data shared

2. The authors may consider revising the order of the sections - possibly moving the "origin of tumor-resident bacteria" to earlier in the manuscript, as this is a question that pops-up almost instantly when reading the manuscript and does not make much sense to leave until near the end of the review. Additionally, greater mechanistic insight could be shared in this section, as it is crucial for setting the precedent of the review. Moreover, it would be helpful for the authors to highlight the currently, or most widely, accepted view of the field.

We agree that discussing the origin of tumor-resident bacteria earlier in the manuscript provides a more logical flow and helps establish a foundational understanding for the rest of the review. In response, we have moved the "Origin of Tumor-Resident Bacteria" section closer to the beginning of the manuscript, after the introduction. Furthermore, we have greatly expanded this section to include greater mechanistic insight into how these bacteria may localize within tumor tissues and we now highlight the currently accepted views in the field regarding their origin and establishment. We believe these changes strengthen the manuscript and improve its clarity and coherence.

3. the manuscript would be better suited to incorporate a few tables, which would concisely organize the information presented. As suggested by the reviewer, we have incorporated three tables into the manuscript to concisely organize and present key information.

Tables summaries to incorporate:

- i. TRB regulation of tumor initiation and progression - bacteria/factor(s) influencing cancer/how it influences cancer and which pathways (pro- or anti-tumor)/signaling involved/references
- ii. TRB in cancer therapy - bacteria/therapy type/influence on therapy/references
- iii. The advantages and limitations of current methods for analyzing TRB

4. Consider adding key points to further address/expand upon in this review: Key aspects of the TME that promote the retention and survival of bacteria. How bacteria can sediment in the high-pressure environment of the tumor and circumvent the immune response in both the blood (if this is indeed the route of translocation) and tumor. Address greater mechanistic detail on pathways that bacteria use to perturb immune cell phenotype, function, and migration in the context of cancer initiation/progression and therapy. Expand on current explanation/observations on organ-specific tropism of TRB.

Thank you for your thoughtful suggestions. As requested, we have expanded the review to address all the key points you mentioned. Specifically, we have:

- Included a discussion on key aspects of the TME that facilitate the retention and survival of bacteria.
- Added greater mechanistic detail on the pathways by which bacteria can influence immune cell phenotype, function, and migration in the context of cancer initiation, progression, and therapy.
- Expanded our discussion on organ-specific tropism of TRB.

Referee #2:

In their review, Vella & Rescigno discuss how tumor-resident bacteria (TRB) influence cancer progression through interactions with the tumor microenvironment, impacting immunity, therapy response, and metastasis. They cover key aspects such as TRB's role in carcinogenesis, effects on antitumoral drugs, immune modulation/cancer immunotherapy and metastatic spread. Additionally, they explore the potential origins of TRB and highlight challenges in the field. Overall, the review is well-written and provides a broad overview of critical questions in the field. However, some important literature is missing, and a few minor revisions should be addressed. A subsection on how TRB research could contribute to cancer prevention would be a valuable addition.

1. Intro

The authors should define 16S rRNA as it appears for the first time in the review. While it is discussed in detail in the challenges section, adding a couple of sentences in the introduction would help non-expert readers. **We thank the reviewer for the helpful suggestion. As requested, we have now defined 16S rRNA upon its first mention in the Introduction section.**

2. TRB and Carcinogenesis

This section lacks depth and does not adequately cover key recent advancements in the field. Several important studies are missing, such as:

- *Fusobacterium nucleatum* (Fn) has been identified in sites beyond colorectal cancer, including breast cancer (Parhi et al., Nat Com, 2020, PMID: 32591509).
- Fn affects cancer stem cells via formate production (Ternes et al., 2022, PMID: 35437333) .
- The modulation of aspirin by Fn and its implications for cancer prevention should be discussed (PMID: 33824205).

We thank the reviewer for pointing out the need to expand the "TRB and carcinogenesis" paragraph and for highlighting these important studies. We have revised the section to provide greater depth and have now included the suggested recent advancements relevant to the role of Fn in carcinogenesis. Also, we have discussed the modulation of aspirin by Fn in cancer prevention.

3. Metastatic Spread

The authors mention the pro-metastatic niche, do they mean pre-metastatic niche? If so, the terminology should be clarified. If they intend to differentiate the two, a discussion on the distinction would be helpful. **We meant pre-metastatic niche. We apologize for the typo**

4. Origin of TRB

- The molecule GalGalNAc should be written in full upon first mention. Additionally, its presence in breast and other cancers should be discussed

As requested, we have now written GalGalNAc in full upon its first mention in the manuscript. Additionally, we have expanded the text to include a discussion of its presence in several cancer types, with appropriate references added.

- The authors could also expand on how comparing oral, fecal, and tumor-resident bacterial strains' genomes might provide insights into TRB origins.

We thank the reviewer for this valuable suggestion. As requested, we have expanded the relevant section to elaborate on how comparing the genomes of oral, fecal, and tumor-resident bacterial strains can provide insights into the origins of TRB. Specifically, we now discuss how 16S and single-nucleotide variant analyses across different body sites can help trace bacterial migration routes.

5. Challenges

-The discussion on 3D culture models should be expanded, addressing potential limitations and technical challenges, such as inverted organoids and injection methods.

We appreciate the reviewer's insightful comment. In response, we have expanded the discussion on 3D culture models to address potential limitations and technical challenges, including issues related to inverted organoid polarity and bacterial injection methods. Also, we have listed limitations and technical challenges.

Minor points:

-In the origin of TRB section, correct the typo: "without casing" → "without causing".
-All bacterial acronyms should follow the convention of introducing the full name first before using abbreviations.

Thank you for pointing this out. We have corrected the typo in the Origin of TRB section, changing "without casing" to "without causing." Additionally, we have revised the manuscript to ensure that all bacterial acronyms are introduced with the full name on first mention, in accordance with standard conventions.

Referee #3:

In this well-written review, the authors explore the complex role of the human microbiota, particularly focusing on tumor-associated bacteria (TRB) and their implications for cancer biology. I think the authors can elaborate more on the mechanisms of Fn in CRC. For instance, it has been shown that Fn expresses amyloid FadA to stimulate CRC growth (Meng et al, EMBO Rep 2021). Another study reported that Fn preferentially binds Annexin A1-expression cells and further stimulates Annexin A1 expression via FadA (Rubinstein et al EMBO Rep 2019). Annexin A1 is upregulated in CRC, which explains why Fn is enriched in CRC. The authors mentioned 5-FU in the review. I want to point out that a previous study reported Annexin A1 expression in CRC caused resistance to 5-FU, consistent with Fn causing resistance to 5-FU (Onozawa et al, Onco Rep 2017). We thank the reviewer for these insightful suggestions and for highlighting key mechanistic studies on Fn in CRC. As recommended, we have expanded the relevant section of the review to include additional details on Fn-mediated mechanisms in CRC. We believe these additions strengthen the review's discussion of Fn's contribution to CRC progression and therapy resistance.

As for origin of the bacteria, there have been several studies reporting Fn identified in CRC originate from the mouth, which is the natural habitat of Fn. Several studies showed that oral gavage of Fn induced tumor formation in Apc min mice (Kostic et al, Cell Host Micro 2013; Rubinstein et al, EMBO Rep 2019). Therefore, Fn may colonize in CRC via multiple pathways.

Additional comments are following:

Abstract Section:

1. In the phrase "tumor-promoting and tumor-suppressing pathways," consider adding a comma before "and" for clarity: "tumor-promoting, and tumor-suppressing pathways." **Thank you for the suggestion. However, we believe that the comma before "and" is not necessary, as the sentence is already clear and flows well without it, following standard grammatical conventions.**
2. The phrase "requires the use of multi-modal technologies" could be clearer. Consider rephrasing to "requires the application of multi-modal technologies." **Thank you, we made the requested changes.**
3. The abstract must include keywords. **Thank you, we have included the requested keywords.**

Introduction Section:

1. In "the developments of next-generation sequencing (NGS) technologies, particularly the 16s rRNA gene sequencing," there should be a space between "the" and "16s."
2. In "the tumor-associated microbiota is now recognized as an integral part of the TME," consider changing "is now recognized" to "has now been recognized" for consistency with the past tense used throughout the article.
3. In "highly intricated and dynamic environment," "intricated" is not commonly used in this context; consider using "intricate" instead.

Thank you, we made the changes requested in points 1-3.

4. Some paragraphs, particularly in the Introduction, are quite long. Consider breaking them into shorter paragraphs for better readability. **Thank you for the suggestion. We tried to find places were to introduce shorter paragraphs, but it was hard to find them.**

Others:

1. Ensure consistent punctuation within citations. For example, there is an inconsistent use of punctuation before citations (such as missing commas) throughout the text. A common format is to place the citation at the end of the sentence before the period.
2. In the sentence** "CagA distrupts various intracellular signaling pathways involved in cell proliferation and apoptosis,"** correct "distrupts" to "disrupts."
3. "promotes of a pro-inflammatory microenvironment conducive to oncogenesis," remove "of." It should read: "promotes a pro-inflammatory microenvironment conducive to oncogenesis." The phrase "enhancing cell proliferation and promoting promoted colonic tumorigenesis" has redundancy with the word "promoted." It should read: "enhancing cell proliferation and promoting colonic tumorigenesis."
4. The phrase "This activation occurs following bacterial binding and colonization of host cells via the bacterial surface protein TMPC, which interacts with the Annexin A2 (ANXA2) receptor on gastric epithelial cells." Consider rephrasing for conciseness: "This activation occurs after bacterial binding and colonization of host cells via the surface protein TMPC, which interacts with the Annexin A2 (ANXA2) receptor on gastric epithelial cells."
5. "In addition to modulating immune responses and signaling pathways, TRB can induce DNA damage," consider specifying "tumor-resident bacteria" (TRB) instead of just "TRB" once at the beginning for clarity.
6. The phrase "Overall, these observations indicate that bacteria contribute to cancer development through various mechanisms, playing a significant and multifaceted role in the malignant transformation of cells," can be more concise: "These observations indicate that bacteria significantly contribute to cancer development through various mechanisms, playing a multifaceted

role in malignant transformation."

7. In the phrase: "Emerging evidences point into the direction," it should be "Emerging evidence points in the direction."

Following the reviewer's suggestion, we have made the changes as requested in points 1-7

8. The sentence "While in vitro *P. gingivalis*-primed human iNKT cells displayed reduced expression of cytotoxic molecules such as granzyme B and perforin and impaired lytic degranulation," could be clearer if rephrased: "In vitro, *P. gingivalis*-primed human iNKT cells exhibited reduced expression of cytotoxic molecules, such as granzyme B and perforin, along with impaired lytic degranulation."

We rephrased the sentence following the reviewer suggestion's and we split the long period into two sentences as follows. "In vitro, *P. gingivalis*-primed human iNKT cells exhibited reduced expression of cytotoxic molecules, such as granzyme B and perforin, along with impaired lytic degranulation. CHI3L1 blockade using a neutralizing antibody restored their killing activity against human CRC cell lines".

9. The phrase "this bacterium induces macrophage infiltration via CCL20-mediated pathways and promotes the polarization of macrophages toward an M2 phenotype, which is associated with a pro-tumoral role" could be streamlined: "This bacterium induces macrophage infiltration via CCL20-mediated pathways and promotes M2 polarization, associated with a pro-tumoral role." We agree in streamlining the phrase above-mentioned.

10. When discussing "bacterial peptides on the surface of cancer cells via the Human Leukocyte Antigens (HLA) class-I and -II molecules," it might be beneficial to briefly explain the significance of this process for readers who may not be familiar with immunology.

We appreciate the reviewer's suggestion and agree that providing additional context would benefit readers who may not have a strong immunology background. Accordingly, we have revised the paragraph to include a brief but informative explanation of the roles of HLA class I and class II molecules in bacterial antigen presentation.

11. In "Colonization of an immunogenic bacterium, *Helicobacter hepaticus*, have been shown to drive induction and maturation of TLSs," change "have" to "has." It should read: "Colonization of an immunogenic bacterium, *Helicobacter hepaticus*, has been shown to drive induction and maturation of TLSs."

12. The sentence "These observations suggest that certain TRB persist throughout the metastatic process, migrating from the primary tumor to metastatic sites, and may play a role in the cancer metastasis process." could be more concise: "These observations suggest that certain TRB persist during metastasis, migrating from primary tumors to metastatic sites, and may actively contribute to the metastatic process."

13. In the phrase "accumulate data indicate that they are not merely passive 'passengers' but active participants," change "accumulate" to "accumulating" for grammatical correctness. Also, consider simplifying to: "data indicate that they are not merely passive 'passengers' but active participants."

14. The phrase "this reprogramming is partly explained by the upregulation of the long non-coding RNA EVADR, which acts as a modular scaffold for Y-box binding protein 1 (YBX1), subsequently enhancing the translation of EMT-associated factors such as Snail, Slug, and Zeb in host cells" is a bit lengthy. Consider breaking it down for clarity or simplify by focusing on the key message: "This reprogramming is linked to the upregulation of long non-coding RNA EVADR, which scaffolds Y-box binding protein 1 (YBX1) to enhance the translation of EMT-associated factors in host cells."

15. The sentence "Studies on CRC mouse models revealed that tumor-resident Escherichia coli C17 can disrupt the gut vascular barrier (GVB) and translocate to the liver via the hematogenous route" could be streamlined: "CRC mouse models revealed that tumor-resident Escherichia coli C17 disrupts the gut vascular barrier (GVB) and translocates to the liver."

16. The sentence "This breach in the mucosal barrier is a potential source of TRB, particularly in tumors originating from mucosal sites with external cavity exposure, such as lung, pancreatic, colorectal, and cervical cancers" could be streamlined: "Compromised mucosal barriers may serve as a source of TRB, especially for tumors from mucosal sites with external cavity exposure, including lung, pancreatic, colorectal, and cervical cancers."

17. In "the administration of engineered probiotic bacteria have emerged as promising approaches", change "have" to "has"

Following the reviewer's suggestion, we have made changes as requested in points 11-17.

18. The phrase "This is due to the very low levels of bacterial DNA typically contained in tumors-several orders of magnitude less than what is found in the gut microbiome-and the overwhelming presence of host genomic DNA, that makes it difficult to distinguish bacteria truly present in the tumor from those introduced through experimental contamination during sample collection and processing" can be streamlined: "This is due to very low levels of bacterial DNA in tumors-several orders of magnitude less than in the gut microbiome-and the overwhelming presence of host genomic DNA, making it difficult to distinguish bacteria genuinely present in the tumor from those introduced through experimental contamination during sample processing." The phrase has been changed in "This is due to very low levels of bacterial DNA in tumors-several orders of magnitude less than in the gut microbiome-and the overwhelming presence of host genomic DNA, making it difficult to distinguish bacteria genuinely present in the tumor from those introduced through experimental contamination during sample collection and processing".

19. In "The prokaryotic 16S rRNA gene is approximately 1500 bp long, with nine variable regions (V1-V9) interspersed with conserved regions," the word "interspersed" can be made clearer by specifying "that are interspersed with conserved regions."

20. In the phrase "fluorescence in situ hybridization (FISH), immunohistochemistry, electron microscopy are commonly used to asses TRB presence and abundance," adjust to "assess" instead of "asses" and add a comma before "and" for correct punctuation: "fluorescence in situ hybridization (FISH), immunohistochemistry, and electron microscopy are commonly used to assess TRB presence and abundance."

Following the reviewer's suggestion, we have made changes as requested in points 19-20.

21. Replace outdated references, such as those from 2000 or 1907, with more current ones.

Thank you for your comment. While we understand the concern regarding older references, we have intentionally retained the citations to Dudgeon & Dunkley (1907) and Bubendorf et al. (2000) because they represent the first documented observations of key phenomena discussed in our review. Specifically, Dudgeon & Dunkley (1907) provided the earliest report of microorganisms observed within tumors, and Bubendorf et al. (2000) offered foundational evidence of breast cancer's preferential metastasis to bone. We believe these references are important for providing historical context and acknowledging the origins of these significant discoveries. To address your suggestion and ensure our references are up-to-date, we have also included more recent studies in

the organotropism section, such as the research paper "The landscape of metastatic progression patterns across major human cancers" (2015) and the review "Metastasis organotropism: redefining the congenial soil" (2019).

22. At the end it would be interesting if you added this point to the manuscript, in light of the challenges detailed in your article regarding the low levels of bacterial DNA in tumor samples, how do you envision future methodological advancements-such as microbial enrichment techniques or sequencing technologies-contributing to a more definitive understanding of the functional roles of TRB in tumor biology and their interactions within the tumor. Thank you for the suggestion. We have added a section discussing how future advancements, such as microbial enrichment techniques and sequencing technologies, could address the challenges of low bacterial DNA levels in tumor samples and enhance our understanding of TRB's functional roles and interactions in tumor biology.

23. The authors drew Fn as curved rod with polar flagellum, which is incorrect. Fn has no flagellum. It is a long filamentous rod with tapered ends, i.e. like a fusiform (thus the name). Thank you, we have made the requested changes.

Prof. Maria Rescigno
Humanitas University
Via Rita Levi Montalcini
Pieve Emanuele 20139
ITALY

Dear Prof. Rescigno,

I am pleased to inform you that your manuscript has been accepted for publication in EMBO reports. Your manuscript will be processed for publication by EMBO Press. It will be copy edited and you will receive page proofs prior to publication. Please note that you will be contacted by Springer Nature Author Services to complete licensing information.

Yours sincerely,
